# Robustify Spiking Neural Networks via Dominant Singular Deflation under Heterogeneous Training Vulnerability

**Desong Zhang, Jia Hu,**\* **and Geyong Min**\*
Department of Computer Science, University of Exeter, Exeter, U.K.
{dz288, J.Hu, G.Min}@exeter.ac.uk

## Abstract

Spiking Neural Networks (SNNs) process information via discrete spikes, enabling them to operate at remarkably low energy levels. However, our experimental observations reveal a striking vulnerability when SNNs are trained using the mainstream method—direct encoding combined with backpropagation through time (BPTT): even a single backward pass on data drawn from a slightly different distribution can lead to catastrophic network collapse. We refer to this phenomenon as the heterogeneous training vulnerability of SNNs. Our theoretical analysis attributes this vulnerability to the repeated inputs inherent in direct encoding and the gradient accumulation characteristic of BPTT, which together produce an exceptional large Hessian spectral radius. To address this challenge, we develop a hyperparameter-free method called **D**ominant **S**ingular **D**eflation (DSD). By orthogonally projecting the dominant singular components of gradients, DSD effectively reduces the Hessian spectral radius, thereby preventing SNNs from settling into sharp minima. Extensive experiments demonstrate that DSD not only mitigates the vulnerability of SNNs under heterogeneous training, but also significantly enhances overall robustness compared to key baselines. Code available at https://github.com/Apple26419/SNN_DSD.

## 1 Introduction

As an emerging brain-inspired computational paradigm, Spiking Neural Networks (SNNs) leverage event-driven, discrete spike streams for feature representation (Maass, 1997). By eliminating the need for pervasive and computationally intensive matrix multiplications of traditional Artificial Neural Networks (ANNs), SNNs achieve remarkable computational efficiency and significantly lower energy consumption (Pei et al., 2019; Meng et al., 2023). Owing to these inherent advantages, SNNs have been applied across a diverse array of application domains, such as autonomous driving (Zhu et al., 2024; Shalumov et al., 2021; Viale et al., 2021), edge computing (Liu et al., 2024a; Zhang et al., 2024), image process (Liu et al., 2025; Pan et al., 2024), and robot control (Jiang et al., 2025).

In the practical deployment of SNNs, safety and reliability are of paramount importance, particularly in terms of robustness against perturbations. Even subtle perturbations in the input data that are imperceptible to human senses can trigger severely adverse and unpredictable network responses (Ding et al., 2024a). To enhance the robustness of SNNs, existing studies predominantly adopt a homogeneous training paradigm, where models are trained on data drawn from a single, uniform distribution—for instance, vanilla training using only clean samples (Ding et al., 2024b;a; Geng & Li, 2023; Ding et al., 2022), or adversarial training where all inputs are perturbed with equal intensity (Ding et al., 2024b; Geng & Li, 2023; Liu et al., 2024b). However, such training settings are idealized and do not reflect the variability and complexity of real-world data. In practical scenarios, models are often required to learn from inherently unpredictable and heterogeneous data distributions, as adversaries may employ a wide range of poisoning strategies to deliberately disrupt distributional homogeneity. We refer to this more realistic paradigm as heterogeneous training (hetero-training). Notably, from the perspective of the attacker, when the number of manipulable samples is limited,

---

\*Corresponding author.

Figure 1: The vulnerability of SNNs in heterogeneous training.

concentrating these perturbed samples as a batch, rather than dispersing them sporadically throughout the dataset, often leads to a more pronounced degradation of model performance (Zou et al., 2022). When exposed to batch-level heterogeneity in the training data, we observe:

**Observation 1.** *In SNN training phase, even a single backward pass with a slightly differently-distributed batch can trigger complete model collapse. As depicted in Fig. 1, SNNs trained on homogeneous datasets—whether comprised solely of clean samples or perturbed ones—exhibit a stable training trajectory. However, introducing just one batch of perturbed data into a clean dataset, or vice versa, leads to abrupt and catastrophic model collapse.* **We refer to this phenomenon as the heterogeneous training vulnerability of SNNs.** *(Sec. 3.1 presents a comprehensive analysis of the experimental results regarding the Observation 1.)*

This observation reveals a fundamental security risk in SNNs when dealing with training data that is inherently unpredictable and cannot be predefined—a scenario often encountered in real-world adversarial contexts (Goodfellow et al., 2014; Kurakin et al., 2018). This prompts these questions:

> **1.** Why do SNNs experience model collapse in hetero-training?
> **2.** Without relying on input data manipulation, how to design an approach for SNNs that effectively mitigates the model collapse induced by hetero-training and enhance robustness?

Motivated by these questions, we propose a novel training method that enhances the robustness of SNNs under both homogeneous and heterogeneous training conditions. Specifically,

- We theoretically show that BPTT yields a Gauss-Newton Hessian with at most linear spectral growth, and that direct encoding makes this bound tight, explaining the abnormally large spectral radius underlying SNN hetero-training vulnerability.
- Building on these theoretical insights, we develop a hyperparameter-free Dominant Singular Deflation (DSD) method. By explicitly deflate the dominant singular components of gradients, DSD effectively reduces the Hessian spectral radius, thereby preventing the network from falling into sharp minima.
- Extensive experimental results demonstrate that DSD mitigates SNN vulnerabilities and significantly enhances robustness under both homogeneous and heterogeneous training conditions, outperforming key baselines and thereby ensuring greater safety in deployment.

## 2 PRELIMINARY

**Spiking neuron dynamic.** In SNNs, neurons emulate the spiking behavior of biological neurons to facilitate information transmission. One of the most prevalent nonlinear spiking neuron models in SNNs is the Leaky Integrate-and-Fire (LIF) neuron (Xu et al., 2022; Fang et al., 2021; Ding et al., 2022). The dynamics of a LIF neuron are described by Eq. (1), where $I_t$, $V_t$, and $S_t$ represent the input current, membrane potential, and spike output at time $t$, respectively. Here, $\tau$ denotes the membrane time constant, $V_{\text{th}}$ is the potential threshold, and $\Theta$ corresponds to the Heaviside function.

$$\tau \frac{dV_t}{dt} = -V_t + I_t, \quad S_t = \Theta(V_t - V_{\text{th}}). \tag{1}$$

**Adversarial attack.** Given an input $x$ with label $y$, adversarial examples are generated by finding a perturbation $\delta$ within an $\ell_p$-norm ball of radius $\epsilon$ that maximizes the loss $\mathcal{L}(h(x + \delta), y)$. This optimization problem is formally expressed as:

$$\underset{\|\delta\|_p \leq \epsilon}{\arg\max} \, \mathcal{L}(f(x + \delta), y). \tag{2}$$

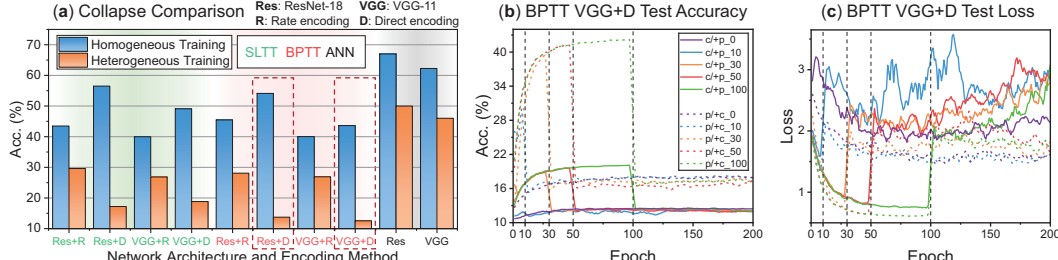

Figure 2: SNN model degradation under hetero-training and collapse curves. In (a), homogeneous training uses clean CIFAR-10 data, while hetero-training employs perturbed CIFAR-10 data. In (b) and (c), the "clean/+perturbation_10 (c/+p_10)" setting denotes homogeneous training on clean CIFAR-10 for the first 9 epochs, followed by hetero-training starting from epoch 10, during which one random batch per epoch is poisoned with perturbed data. Conversely, the "p/+c" setting denotes homogeneous training on perturbed data, with the heterogeneous phase poisoning one random batch per epoch using clean data. All perturbations are generated with FGSM on CIFAR-10 using $\epsilon = 2$.

In this paper, we employ two widely adopted adversarial attack methods in main experiments—Fast Gradient Sign Method (FGSM) (Goodfellow et al., 2014) and Projected Gradient Descent (PGD) (Madry et al., 2017). Hyperparameter configurations are provided in Appendix A.

## 3 ANALYSIS AND METHOD

In this section, we first present experimental results that demonstrate the model collapse phenomenon described in *Observation 1* and analyze why SNNs exhibit a disconcerting vulnerability in hetero-training (Sec. 3.1). Building on this analysis, we introduce the Dominant Singular Deflation method and provide a theoretical analysis explaining how our approach mitigates the vulnerabilities associated with hetero-training while simultaneously enhancing network robustness (Sec. 3.2).

### 3.1 PRELIMINARY EXPERIMENT: WHY SNN MODEL COLLAPSES IN HETERO-TRAINING?

We conduct controlled experiments to examine network collapse in SNNs under heterogeneous training across three core factors: training paradigm, encoding method, and network architecture. For training paradigms, we consider BPTT and SLTT (Spatial Learning Through Time (Meng et al., 2023)). BPTT backpropagates gradients through temporal multiplication, whereas SLTT eliminates the multiplicative terms in BPTT. For encoding methods, we evaluate direct encoding and rate encoding (Poisson). For network architectures, we adopt two widely used backbones: ResNet-18 and VGG-11. Further experimental settings are provided in Appendix B.1. Fig. 2(a) presents all combinations of these three factors, alongside ANN baselines, and compares their collapse behaviors under hetero-training. The most severe collapses: BPTT+ResNet-18+direct encoding and BPTT+VGG-11+direct encoding, are highlighted with red boxes, and both collapse to the point of exhibiting almost no effective training. From these comparisons, we draw the following observations:

**Observation 2.** *(i). Under hetero-training, ANNs exhibit moderate degradation yet far from collapse, SLTT suffers much less collapse compared with BPTT. (ii). Direct encoding leads to far more severe collapse than rate encoding. (iii). Regarding architectures, both ResNet-18 and VGG-11 show comparable levels of degradation or collapse, suggesting that collapse is not tied to architecture.*

To further illustrate this phenomenon, Fig. 2(b) and (c) display training curves of the BPTT+VGG-11+Direct Encoding combination when switching abruptly from homogeneous to heterogeneous training at different stages. In all cases, such a switch induces catastrophic collapse, manifested by an immediate drop in accuracy, a complete degradation in loss, and unstable oscillations thereafter. Taken together, these results suggest that the damage caused by hetero-training is independent of network architecture and training stage, but strongly dependent on the training paradigm (BPTT) and the encoding method (direct encoding).

This phenomenon suggests that, within a single epoch, the BPTT+direct encoding combination may intermittently drive the network parameters into extremely sharp local minima, characterized by an

abnormally large spectral radius of the loss Hessian (Cheng et al., 2022). To interpret this behavior, we conduct an analysis in the context of SNN-specific properties as follows.

**Theorem 1** (Layer-wise GN spectral bound under BPTT with LIF neuron dynamics). *For the $l$-th layer parameters $W^l$, the BPTT gradient expansion (Xiao et al., 2022; Huang et al., 2024) is*

$$\frac{\partial L}{\partial W^l} = \sum_{t=1}^{T} \left[ \underbrace{\frac{\partial L}{\partial S_t^{l+1}} \frac{\partial S_t^{l+1}}{\partial V_t^{l+1}}}_{G_t} \overbrace{\left( \underbrace{\frac{\partial V_t^{l+1}}{\partial W^l}}_{D_t} + \sum_{k<t} \prod_{i=k}^{t-1} \underbrace{\left( \frac{\partial V_{i+1}^{l+1}}{\partial V_i^{l+1}} + \frac{\partial V_{i+1}^{l+1}}{\partial S_i^{l+1}} \frac{\partial S_i^{l+1}}{\partial V_i^{l+1}} \right)}_{J_{k:t-1}} \frac{\partial V_k^{l+1}}{\partial W^l} \right)}^{g_t} \right]. \quad (3)$$

*By bounded surrogate derivatives and the contractive LIF dynamics (See Appendix C for detailed derivations for all bounds mentioned in this analysis), there exist constants $C_G, C_D < \infty$ and $\rho \in (0,1)$, independent of $t$ and $T$, such that $\|G_t\| \leq C_G$, $\|D_t\| \leq C_D$, and $\left\| \frac{\partial V_{i+1}^{l+1}}{\partial V_i^{l+1}} + \frac{\partial V_{i+1}^{l+1}}{\partial S_i^{l+1}} \frac{\partial S_i^{l+1}}{\partial V_i^{l+1}} \right\| \leq \rho$. Then, we have $\|J_t^W\| = \left\| D_t + \sum_{k<t} J_{k:t-1} D_k \right\| \leq \frac{C_D}{1-\rho} = C_J$, therefore $\|g_t\| = \|G_t J_t^W\| \leq C_G C_J$. Hence each per-step gradient contribution is $O(1)$. The Gauss–Newton (GN) Hessian block with respect to $W^l$ satisfies*

$$H(W^l) \approx \sum_{t=1}^{T} (J_t^W)^\top H_t J_t^W \succeq 0, \quad (4)$$

*where $H_t = B_t^\top H_{z,t} B_t$ is the effective Hessian with respect to the membrane potential $V_t^{l+1}$, $H_{z,t}$ indicates the output-layer Hessian at time $t$, and $B_t = \frac{\partial z_t}{\partial V_t^{l+1}}$ denotes the readout Jacobian from the membrane potential to the output logits $z_t$. Since $H_{z,t} \succeq 0$ and $\|B_t\| \leq C_B$, where $C_z = \sup_t \lambda_{\max}(H_{z,t}) < \infty$, we have $\|H_t\| \leq C_B^2 C_z$, and therefore*

$$\lambda_{\max}(H(W^l)) \leq C_B^2 C_z \sum_{t=1}^{T} \|J_t^W\|^2 \leq \underbrace{C_B^2 C_z C_J^2}_{constant} T, \quad (5)$$

*Thus, the largest eigenvalue (equivalently, the spectral radius) of $H(W^l)$ is linearly bounded in $T$.*

**Theorem 2** (Direct encoding makes the GN bound tight). *Consider the same setting as Theorem 1 with the GN block $H(W^l) \approx \sum_{t=1}^{T} (J_t^W)^\top H_t J_t^W \succeq 0$, where $H_t = B_t^\top H_{z,t} B_t$. Under direct encoding, the per-step inputs are stationary, and the recurrent Jacobians become nearly time-invariant up to bounded perturbations (Zenke & Ganguli, 2018; Bellec et al., 2020). Consequently, by the power-iteration effect of repeatedly applying contractive operators, the BPTT Jacobians $\{J_t^W\}$ concentrate along a common dominant singular direction. Formally, there exist unit vectors $a$ (output space) and $b$ (parameter space), scalars $\alpha_t$, and residual terms $R_t$ such that for all $t$ we have*

$$J_t^W = \alpha_t ab^\top + R_t, \qquad m \leq |\alpha_t| \leq M, \qquad \sum_{t=1}^{T} \|R_t\|^2 = o(T), \quad (6)$$

*where $m, M > 0$ denote finite constants independent of $t$ and $T$, and we use the Landau notation $o(T)$ to denote a sublinear term, i.e., $f(T) = o(T)$ if $f(T)/T \to 0$ as $T \to \infty$. Moreover, suppose the output-layer curvature along $a$ has a strictly positive time-average, $\frac{1}{T} \sum_{t=1}^{T} a^\top H_t a \geq c_z^- > 0$, where $c_z^-$ denotes a uniform positive lower bound, serving as the lower-bound counterpart of $C_z$. Then we have equation as follows, where $\Theta(\cdot)$ denotes the standard asymptotic order notation.*

$$\lambda_{\max}(H(W^l)) = \Theta\left( \sum_{t=1}^{T} \alpha_t^2 \right) = \Theta(T). \quad (7)$$

Together, Theorem 1 and Theorem 2 establish a two-stage picture of the Gauss–Newton curvature in SNNs. The first result shows that under LIF dynamics the per-step BPTT contributions are uniformly bounded, and consequently the spectral radius of the Hessian grows at most linearly in $T$. The second

result demonstrates that when direct encoding is used, the stationarity of inputs and the time-invariant structure of the recurrent operators lead to a power-iteration effect, aligning the Jacobians $J_t^W$ along a common rank-one component. This alignment ensures that the lower bound grows at the same linear rate, thereby making the Gauss–Newton bound asymptotically tight. In summary, BPTT establishes the $O(T)$ upper bound for spectral radius, and direct encoding sharpens it to $\Theta(T)$. This spectral pathology becomes particularly severe under heterogeneous training, where even a small fraction of distributional variation is sufficient to excite the sharp directions amplified by the $O(T)$ curvature growth. Because the dominant Hessian eigenmodes scale linearly with $T$, perturbations from mismatched batches accumulate disproportionately along these fragile directions, causing instabilities that can quickly lead to collapse. Hence, the vulnerability of SNNs under hetero-training can be traced to the same mechanism identified above: the alignment of BPTT Jacobians and the resulting unbounded growth of the spectral radius.

Furthermore, we conduct experiments to validate our theoretical analysis. As shown in Fig. 3, we train on CIFAR-10 with the VGG-11 architecture under direct encoding at different time step scales for 200 epochs (additional experimental details are provided in Appendix B.1). The results indicate that network degradation under hetero-training becomes increasingly severe as the number of time steps grows. This confirms that time step scaling is indeed one of the factors affecting robustness, thereby supporting the reliability of our earlier analysis. However, robustness against hetero-training cannot

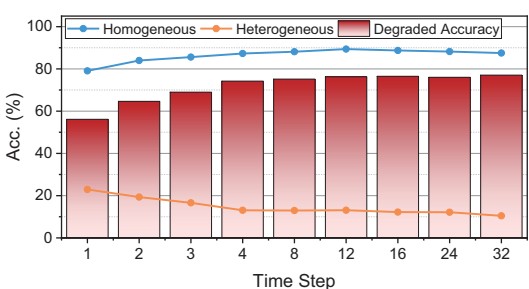

Figure 3: Degradation comparison.

be achieved simply by reducing the number of time steps, since doing so causes a drastic performance drop under homogeneous training as Fig. 3. This motivates the need for a mechanism that can actively suppress the dominant curvature growth, thereby enhancing the robustness and hetero-training resistance without sacrificing the efficiency of direct encoding.

## 3.2 REDUCING SPECTRAL RADIUS VIA DOMINANT SINGULAR DEFLATION

**Dominant Singular Deflation.** To mitigate the pathological spectral growth identified above, that is, to suppress the excessive enlargement of the Hessian spectral radius, we propose a deterministic and hyperparameter-free gradient update technique named Dominant Singular Deflation (DSD). The key idea is to deflate the gradient by explicitly removing its rank-one dominant singular component, thereby directly reducing the maximal singular value that drives curvature amplification.

Formally, for parameter set $\theta$, let $\nabla_\theta \mathcal{L}(\theta)$ denote the gradient of the loss function, represented as a $k$-dimensional tensor $\nabla_\theta \mathcal{L}(\theta) \in \mathbb{R}^{d_1 \times d_2 \times \cdots \times d_k}$. To systematically analyze its principal components, we introduce a deterministic matrixization operator $\mathcal{M} : \mathbb{R}^{d_1 \times \cdots \times d_k} \to \mathbb{R}^{m \times n}$ with $m = d_1$ and $n = \prod_{j=2}^{k} d_j$. Applying singular value decomposition gives

$$\mathcal{M}\big(\nabla_\theta \mathcal{L}(\theta)\big) = U\Sigma V^\top = \sum_{i=1}^{r} \sigma_i u_i v_i^\top, \quad r = \min(m, n), \quad \sigma_1 \geq \sigma_2 \geq \cdots \geq \sigma_r \geq 0. \quad (8)$$

We refer to $\sigma_1 u_1 v_1^\top$ as the dominant singular component. To remove this component, we define the projection operator $\mathcal{D}(A)$ for any $A \in \mathbb{R}^{m \times n}$ as

$$\mathcal{D}(A) = \frac{\langle A, \ u_1 v_1^\top \rangle_F}{\|u_1 v_1^\top\|_F^2} \ u_1 v_1^\top, \quad (9)$$

where $\langle A, B \rangle_F = \sum_{i,j} A_{ij} B_{ij}$ is the Frobenius inner product. Thus, DSD orthogonally projects $\mathcal{M}\big(\nabla_\theta \mathcal{L}(\theta)\big)$ onto the complement of the dominant singular component, yielding the deflated update

$$\widetilde{\nabla_\theta \mathcal{L}}(\theta) = \mathcal{M}^{-1}\big(\mathcal{M}\big(\nabla_\theta \mathcal{L}(\theta)\big) - \mathcal{D}\big(\mathcal{M}\big(\nabla_\theta \mathcal{L}(\theta)\big)\big)\big). \quad (10)$$

**Effectiveness of DSD.** Under direct encoding, Theorem 2 shows that the GN spectral bound becomes tight: the largest eigenvalue $\lambda_{\max}(H(W^l))$ is governed by the squared maximal singular value of the

Jacobians, i.e., $\lambda_{\max}\big(H(W^l)\big) = \Theta\big(\sum_{t=1}^T \sigma_{\max}(J_t^W)^2\big)$. Hence the curvature growth is dominated by the rank-one component associated with the leading singular pair $(\sigma_1, u_1, v_1)$. By construction, DSD removes this rank-one component $\sigma_1 u_1 v_1^\top$ from the gradient, yielding deflated Jacobians $J_t'$. Consequently, their maximal singular value satisfies $\sigma_{\max}(J_t') = \sigma_2(J_t) < \sigma_1(J_t)$, and the Hessian spectral radius strictly decreases:

$$\lambda_{\max}\big(H(W^l; \widetilde{\nabla_\theta \mathcal{L}(\theta)})\big) < \lambda_{\max}\big(H(W^l; \nabla_\theta \mathcal{L}(\theta))\big). \tag{11}$$

In other words, under the alignment effect induced by direct encoding, DSD deterministically suppresses the spectral radius by eliminating its dominant contributor, thereby mitigating the instability that causes vulnerability in SNN training. Besides, a reduction in Hessian spectral radius can indirectly lower the upper bound of the network's Lipschitz constant (Nesterov, 2013; Yao et al., 2020; Ghorbani et al., 2019), thereby contributing to improved robustness and enhanced generalization capabilities (Ding et al., 2022).

**Descent preservation of DSD.** Although DSD explicitly modifies the gradient by removing its dominant singular component, it does not compromise the descent property of gradient-based optimization. Formally, DSD in Eq. (10) rewrites the matrixized gradient as

$$\mathcal{M}(\widetilde{\nabla_\theta \mathcal{L}(\theta)}) = \mathcal{M}(\nabla_\theta \mathcal{L}(\theta)) - \mathcal{D}\big(\mathcal{M}(\nabla_\theta \mathcal{L}(\theta))\big). \tag{12}$$

The directional derivative of the loss $D\mathcal{L}(\theta)[d]$, i.e., the rate of change of $\mathcal{L}$ at $\theta$ along a direction $d$, is given by the inner product between the gradient and $d$. Under DSD, for the update direction $d = -\widetilde{\nabla_\theta \mathcal{L}(\theta)}$ we obtain

$$D\mathcal{L}(\theta)[d] = \langle \nabla_\theta \mathcal{L}(\theta), d \rangle = -\Big\langle \mathcal{M}(\nabla_\theta \mathcal{L}(\theta)), \mathcal{M}(\widetilde{\nabla_\theta \mathcal{L}(\theta)}) \Big\rangle_F. \tag{13}$$

Since $\mathcal{D}$ is an orthogonal projection operator, we can expand and simplify as

$$\begin{aligned}
\Big\langle \mathcal{M}(\nabla_\theta \mathcal{L}(\theta)), \mathcal{M}(\widetilde{\nabla_\theta \mathcal{L}(\theta)}) \Big\rangle_F &= \big\langle \mathcal{M}(\nabla_\theta \mathcal{L}(\theta)), \mathcal{M}(\nabla_\theta \mathcal{L}(\theta)) - \mathcal{D}\big(\mathcal{M}(\nabla_\theta \mathcal{L}(\theta))\big) \big\rangle_F \\
&\overset{*}{=} \big\| \mathcal{M}(\nabla_\theta \mathcal{L}(\theta)) \big\|_F^2 - \big\| \mathcal{D}\big(\mathcal{M}(\nabla_\theta \mathcal{L}(\theta))\big) \big\|_F^2 \\
&= \big\| \mathcal{M}(\widetilde{\nabla_\theta \mathcal{L}(\theta)}) \big\|_F^2,
\end{aligned} \tag{14}$$

where $(*)$ uses the *self-adjointness* and *idempotence* of the orthogonal projection $\mathcal{D}$ with respect to the Frobenius inner product, namely, $\langle A, \mathcal{D}(A) \rangle_F = \langle \mathcal{D}(A), \mathcal{D}(A) \rangle_F = \|\mathcal{D}(A)\|_F^2$. Substituting this into the directional derivative gives Eq. (15), with strict inequality whenever $\widetilde{\nabla_\theta \mathcal{L}(\theta)} \neq 0$.

$$D\mathcal{L}(\theta)[d] = -\big\| \mathcal{M}(\widetilde{\nabla_\theta \mathcal{L}(\theta)}) \big\|_F^2 \leq 0, \tag{15}$$

In summary, DSD guarantees that the update direction always yields non-increasing loss, and it is a strict descent direction whenever the deflated gradient is non-zero.

**Remark 1.** *This conclusion follows solely from the general mathematical property of orthogonal projections in Hilbert spaces (Horn & Johnson, 2012), and does not rely on any special structure of SNNs, BPTT, or the cross-entropy loss. Hence the guarantee of descent preservation is universal and independent of the particular network architecture or loss function.*

## 4 EXPERIMENT

Our experiments are structured into four parts. First, we assess the robustness of DSD in homogeneous training (Sec. 4.1), which are divided into two settings: vanilla training using clean data and adversarial training (AT) (Kundu et al., 2021) using perturbed data generated by white box FGSM with an $\epsilon$ of $2/255$. Second, we investigate whether DSD can prevent network collapse in hetero-training (Sec. 4.2). Third, we evaluate the effect of our approach on the Hessian eigenvalue during inference (Sec. 4.3). Finally, we inspect DSD for any instances of gradient obfuscation (Sec. 4.4). In addition, the extra computational overhead of DSD during training is reported in Appendix E, while **DSD introduces no overhead at inference**.

To ensure comprehensive evaluation, we conduct experiments on static visual datasets of varying scales, including CIFAR-10 (Krizhevsky et al., 2009), CIFAR-100 (Krizhevsky et al., 2009), TinyImageNet (Deng et al., 2009), and ImageNet (Deng et al., 2009) and Dynamic Vision Sensor (DVS) datasets DVS-CIFAR10 (Li et al., 2017) and DVS-Gesture (Amir et al., 2017). Implementation specifics are provided in Appendix B.2.

## 4.1 COMPARISON WITH STATE-OF-THE-ART (SOTA) IN HOMOGENEOUS TRAINING

**White box attack in static datasets.** Table 1 summarizes DSD accuracies under various homogeneous training settings, compared with SOTA defenses (StoG (Ding et al., 2024b), DLIF (Ding et al., 2024a), HoSNN (Geng & Li, 2023), and FEEL (Xu et al., 2024)). Overall, DSD consistently outperforms prior defenses against gradient-based white box attacks across all datasets and training modes. Notably, it yields over 10% accuracy gains in several cases, including CIFAR-100 with FGSM in vanilla training, CIFAR-10 and ImageNet with FGSM, and TinyImageNet with PGD in AT. In TinyImageNet AT, DSD reaches 30.87% accuracy, a striking 22.68% improvement over the baseline SNN's 8.19%. Robustness under vanilla training remains difficult, particularly against PGD. Even so, DSD achieves 8.09% accuracy on CIFAR-100 under PGD in vanilla training, a major improvement over the previous best of 2.04% from FEEL (Xu et al., 2024).

Table 1: White box performance comparison (%). The highest accuracy in each column is highlighted in bold. The "Improvement" quantifies the gain of DSD over the other best-performing baseline.

| Methods | CIFAR-10 | | | CIFAR-100 | | | TinyImageNet | | | ImageNet | | |
|---|---|---|---|---|---|---|---|---|---|---|---|---|
| | Clean | FGSM | PGD | Clean | FGSM | PGD | Clean | FGSM | PGD | Clean | FGSM | PGD |
| *Homogeneous Training: Vanilla Training* | | | | | | | | | | | | |
| SNN | **93.75** | 8.19 | 0.03 | 72.39 | 4.55 | 0.19 | **56.82** | 3.51 | 0.14 | **57.84** | 4.99 | 0.01 |
| StoG (Ding et al., 2024b) | 91.64 | 16.22 | 0.28 | 70.44 | 8.27 | 0.49 | - | - | - | - | - | - |
| DLIF (Ding et al., 2024a) | 92.22 | 13.24 | 0.09 | 70.79 | 6.95 | 0.08 | - | - | - | - | - | - |
| HoSNN (Geng & Li, 2023) | 92.43 | 54.76 | 15.32 | 71.98 | 13.48 | 0.19 | - | - | - | - | - | - |
| FEEL (Xu et al., 2024) | 93.29 | 44.96 | 28.35 | **73.79** | 9.60 | 2.04 | 43.83 | 9.59 | 4.53 | - | - | - |
| DSD (Ours) | 90.21 | **55.86** | **31.44** | 70.26 | **23.81** | **8.09** | 54.54 | **19.50** | **12.02** | 53.47 | **14.69** | **4.30** |
| Improvement | ▼ 3.54 | ▲ 1.10 | ▲ 3.09 | ▼ 3.53 | ▲ 10.33 | ▲ 6.05 | ▼ 2.28 | ▲ 9.91 | ▲ 7.49 | ▼ 4.37 | ▲ 9.70 | ▲ 4.29 |
| *Homogeneous Training: Adversarial Training (AT)* | | | | | | | | | | | | |
| SNN | **91.16** | 38.20 | 14.07 | 69.69 | 16.31 | 8.49 | **49.91** | 8.19 | 2.97 | **51.00** | 15.74 | 6.39 |
| StoG (Ding et al., 2024b) | 90.13 | 45.74 | 27.74 | 66.37 | 24.45 | 14.42 | - | - | - | - | - | - |
| DLIF (Ding et al., 2024a) | 88.91 | 56.71 | 40.30 | 66.33 | 36.83 | 24.25 | - | - | - | - | - | - |
| HoSNN (Geng & Li, 2023) | 90.00 | 63.98 | 43.33 | 64.64 | 26.97 | 16.66 | - | - | - | - | - | - |
| FEEL (Xu et al., 2024) | - | - | - | **69.79** | 18.67 | 11.07 | - | - | - | - | - | - |
| DSD (Ours) | 86.62 | **74.43** | **44.38** | 64.21 | **43.91** | **27.11** | 46.30 | **30.87** | **18.21** | 49.78 | **26.83** | **9.12** |
| Improvement | ▼ 4.54 | ▲ 10.45 | ▲ 1.05 | ▼ 5.58 | ▲ 7.08 | ▲ 2.86 | ▼ 3.61 | ▲ 22.68 | ▲ 15.24 | ▼ 1.22 | ▲ 10.09 | ▲ 2.73 |

**Stronger white box attack.** Beyond conventional white-box attacks, we further evaluate whether DSD can defend against stronger adversaries. We employ the APGD attack (Croce & Hein, 2020), which incorporates adaptive step-size control and a momentum-like update during iterations, enabling more efficient exploration of the loss landscape and avoiding local optima to generate stronger adversarial samples. Two variants of APGD are used—APGDCE and APGDDLR—with detailed descriptions and hyperparameter settings provided in Appendix A.

Table 2: APGD performance (%). The highest accuracy in each column is highlighted in bold.

| Methods | CIFAR10 | | CIFAR-100 | |
|---|---|---|---|---|
| | APGDCE | APGDDLR | APGDCE | APGDDLR |
| *Homogeneous Training: Vanilla Training* | | | | |
| SNN | 0.61 | 2.06 | 0.09 | 0.12 |
| DLIF | 0.05 | 0.03 | 0.02 | 0.18 |
| HoSNN | 10.35 | 27.39 | 2.55 | 0.02 |
| DSD (Ours) | **22.19** | **29.77** | **5.99** | **5.98** |
| *Homogeneous Training: Adversarial Training (AT)* | | | | |
| SNN | 10.98 | 17.72 | 5.87 | 6.20 |
| DLIF | 35.09 | 39.85 | 20.68 | **24.21** |
| HoSNN | 38.89 | 37.94 | 12.55 | 13.66 |
| DSD (Ours) | **47.08** | **44.62** | **22.86** | 23.04 |

As shown in Table 2, our method achieves the highest accuracy under all attack settings except for APGDDLR on CIFAR-100 with AT, where it slightly lags behind DLIF (Ding et al., 2024a). These results demonstrate that DSD remains effective and superior even against stronger white box attacks.

**White box attack in DVS datasets.** In DVS datasets, we trained the model and performed inference under FGSM and PGD attacks by directly perturbing the preprocessed event frames, as implemented in (Liu et al., 2024b). It can be seen from Table 4 that our method also demonstrates excellent robustness when dealing with the DVS dataset, surpassing SOTA method SR.

Table 3: Performance comparion in DVS datasets (%). he highest accuracy in each column is highlighted in bold. SR: (Liu et al., 2024b)

| Methods | DVS-CIFAR10 | | | DVS-Gesture | | |
|---|---|---|---|---|---|---|
| | Clean | FGSM | PGD | Clean | FGSM | PGD |
| SNN | **76.30** | 17.20 | 5.00 | **95.49** | 39.24 | 9.72 |
| SR | 75.50 | 64.60 | **61.20** | - | - | - |
| DSD | 75.10 | **65.80** | **61.20** | 93.75 | **90.28** | **55.21** |

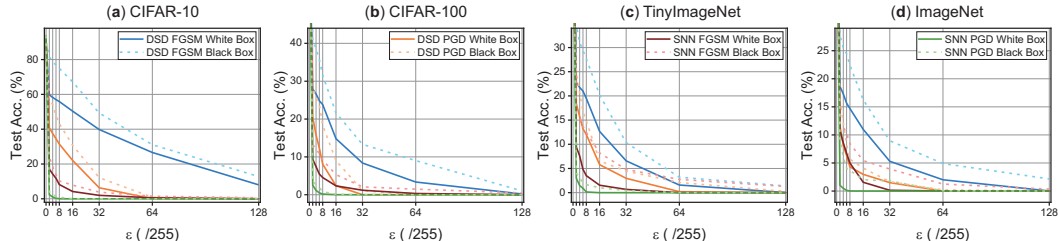

Figure 4: Performance comparison in different white box and black box attacks.

**Black box attack.** Black box adversarial examples are generated using the substitute-model approach in this experiment. We evaluated DSD's resilience across a range of perturbation magnitudes and compared its performance to that of a vanilla SNN under the same attack strengths. Fig. 4 visualizes these results, and the complete experimental data are provided in Appendix F. Across all four dashed-line baselines, DSD consistently achieves substantially higher accuracy than the vanilla model, demonstrating its effectiveness against diverse black box threat scenarios.

## 4.2 PERFORMANCE IN HETEROGENEOUS TRAINING

**Comparison with SOTA methods.** Building on the effectiveness of DSD under homogeneous training, we further examine its behavior under heterogeneous training conditions. We compare DSD against SOTA methods, including RAT (Ding et al., 2022), DLIF (Ding et al., 2024a), and FEEL (Xu et al., 2024). The results are represented by float bar figures as Fig. 5, with detailed numerical values provided in Appendix F. As illustrated in the figure, DSD exhibits the least performance degradation under hetero-training (i.e., it produces the shortest floating bars), and moreover achieves clearly higher absolute accuracy than all competing methods under batch injections with $b = 1$. These findings demonstrate that DSD effectively mitigates the model collapse induced by hetero-training, outperforming existing approaches.

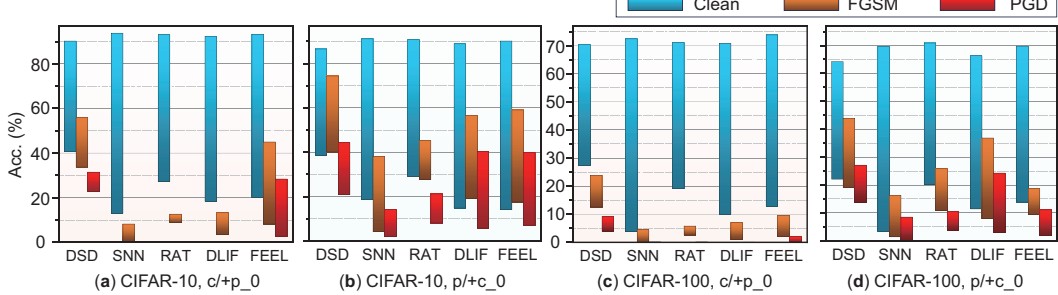

Figure 5: Performance degradation comparison in hetero-training. Following the protocol of Sec. 3.1, we define two poisoning schemes: c/+p_0 and p/+c_0. In c/+p_0, training with clean data, beginning at epoch 0, we inject $b = 0, 1$ FGSM–perturbed ($\epsilon = 2/255$) batches at the end of each epoch; p/+c_0 is defined analogously: trained with perturbed data and injected with clean data. Subfigures (**a**), (**b**) and (**c**), (**d**) are vertically aligned, with each pair sharing the same y-axis. For any floating bar in the figure, the top represents the accuracy at ($b = 0$), the bottom represents the accuracy at ($b = 1$), and the bar length indicates the degree of performance degradation. For some methods under PGD, the floating bars are barely visible because their accuracies at both (b=0) and (b=1) are nearly zero.

**DSD's resilience in different heterogeneous intensities.** Furthermore, we evaluate the maximum poisoning intensity that DSD can tolerate under hetero-training. In this experiment, we inject different numbers of heterogeneous batches ($b = 1, 2, 5$) to measure the extent of performance degradation. The results, including the performance degeneration, visualized in Fig. 6, with detailed experimental data in Appendix F, omit the vanilla SNN baselines due to their near-total failure under hetero-training. Remarkably, DSD maintains strong resilience across all settings, with no instance of full collapse. Even at the highest poisoning strength ($b = 5$), DSD sustains a 30% accuracy on CIFAR-10 under FGSM inference, demonstrating that DSD can withstand high-intensity hetero-training and further indicates its ability to remain robust under realistic batch-level data poisoning scenarios.

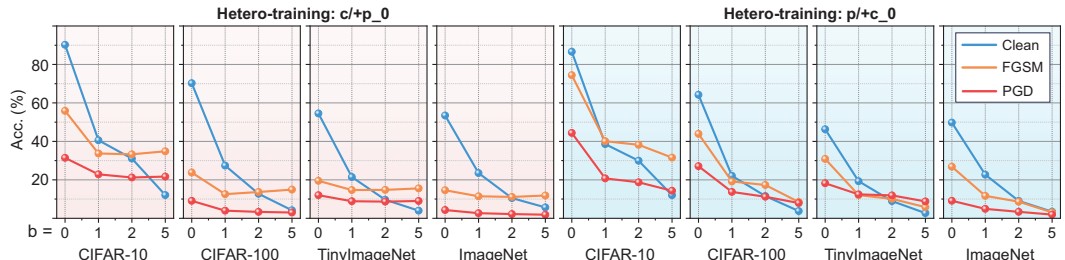

Figure 6: DSD performance in hetero-training. All subfigures share the same y-axis.

## 4.3 HESSIAN EIGENVALUE EVALUATION

In this experiment, we compare the Hessian eigenvalue of the DSD-trained model against those of a vanilla SNN during inference. Specifically, for each inference batch, we compute two metrics: (i). $\lambda_{\max}(H)$: The spectral radius of the Hessian; (ii). $\Pr(H)$: The proportion of $\lambda_{\max}(H)$ within the top-five eigenvalues, serving as an indicator of the overall smoothness of the loss landscape. The details for this experiment are provided in Appendix B.2. Table 4 reports both metrics under three distinct adversarial attack scenarios. Across all cases, DSD consistently achieves a lower $\lambda_{\max}(H)$ and markedly reduces its proportional presence among the top-five eigenvalues. These findings corroborate our theoretical design, demonstrating that DSD indeed reduces the Hessian spectral radius, smooths the Hessian sharpness, and underpins its robustness enhancements.

Table 4: Hessian eigenvalue evaluation. The smaller $\lambda_{\max}(H)$ and $\Pr(H)$ are, the better.

| Methods | CIFAR-10 | | CIFAR-100 | | TinyImageNet | | ImageNet | |
|---|---|---|---|---|---|---|---|---|
| | $\lambda_{\max}(H)$ | $\Pr(H)$ | $\lambda_{\max}(H)$ | $\Pr(H)$ | $\lambda_{\max}(H)$ | $\Pr(H)$ | $\lambda_{\max}(H)$ | $\Pr(H)$ |
| **Clean Inference** | | | | | | | | |
| SNN | 261.94 | 0.98 | 187.49 | 0.41 | 2077.09 | 0.60 | 174.88 | 0.43 |
| DSD | 209.90 ▼52.04 | 0.35 ▼0.63 | 135.55 ▼51.94 | 0.30 ▼0.11 | 1802.80 ▼274.29 | 0.51 ▼0.09 | 110.56 ▼64.32 | 0.33 ▼0.10 |
| **FGSM Inference** | | | | | | | | |
| SNN | 269.90 | 1.15 | 190.82 | 0.46 | 1998.11 | 0.61 | 162.57 | 0.44 |
| DSD | 218.27 ▼51.63 | 0.38 ▼0.77 | 132.01 ▼58.81 | 0.29 ▼0.17 | 1767.57 ▼230.54 | 0.50 ▼0.11 | 111.77 ▼50.80 | 0.32 ▼0.12 |
| **PGD Inference** | | | | | | | | |
| SNN | 265.47 | 1.03 | 200.11 | 0.46 | 2072.77 | 0.62 | 174.16 | 0.41 |
| DSD | 202.89 ▼62.58 | 0.35 ▼0.68 | 143.07 ▼57.04 | 0.30 ▼0.16 | 1793.12 ▼279.65 | 0.56 ▼0.06 | 113.73 ▼60.43 | 0.32 ▼0.09 |

## 4.4 INSPECTION OF GRADIENT OBFUSCATION

Next, we examine whether DSD suffers from gradient obfuscation (Athalye et al., 2018). For items (1) and (2) in Table 5, Fig. 3 presents DSD's accuracy under FGSM and PGD attacks across a range of perturbation bounds, as well as a side-by-side comparison of white box versus black box performance. It is clear that DSD is consistently more vulnerable to iterative PGD than to single-step FGSM, and that white box attacks inflict greater degradation than black box attacks. Items (3) and (4) are also evident in Fig. 3: as the perturbation limit increases, DSD's accuracy drops sharply, even reaching 0% under several settings. Fig. 7 further corroborates this trend, showing that although DSD's performance progressively worsens with more PGD iterations, it eventually converges to a steady minimum. Item (5) indicates that gradient-based attacks fail to locate adversarial examples; however, our results in Fig. 3 demonstrate the opposite—both FGSM and PGD continue to fool DSD despite the training. In short, DSD does not utilize gradient obfuscation to achieve false robustness.

Table 5: Checklist for identifying gradient obfuscation.

| Characteristics to identify gradient obfuscation | Pass? |
|---|---|
| (1) Single-step attack performs better compared to iterative attacks | ✓ |
| (2) Black-box attacks perform better compared to white-box attacks | ✓ |
| (3) Increasing perturbation bound can't increase attack strength | ✓ |
| (4) Unbounded attacks can't reach 100% success | ✓ |
| (5) Adversarial example can be found through random sampling | ✓ |

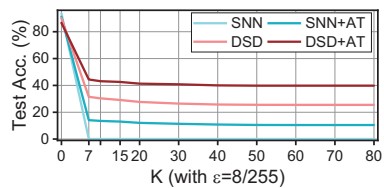

Figure 7: Acc. under different $K$.

## 5 RELATED WORK

**SNN defensive methods.** Direct encoding with BPTT is the prevailing paradigm for SNN training (Wu et al., 2018; Deng et al., 2022; Meng et al., 2023; Xiao et al., 2022; Wang et al., 2023), which improves efficiency but sacrifices the robustness inherent to rate encoding. To counteract this vulnerability, prior studies have explored input noise injection (Kundu et al., 2021), adversarial training with Lipschitz regularization (Ding et al., 2022), gradient sparsity regularization (Liu et al., 2024b), stochastic gating mechanisms (Ding et al., 2024b), randomized smoothing coding (Wu et al., 2024), noise sensitivity regularization (Zheng et al., 2023), adaptive leak dynamics (Xu et al., 2024), and modified objectives to suppress membrane potential perturbations (Ding et al., 2024a). While these methods improve robustness empirically, they do not address the underlying mechanism by which direct encoding with BPTT induces vulnerability.

**Beyond SNNs.** Related ideas of suppressing dominant directions have been studied in the broader context of SNNs. Classical works control spectral growth through Parseval networks (Cisse et al., 2017) or spectral norm regularization (Yoshida & Miyato, 2017), while Lipschitz-margin training (Tsuzuku et al., 2018) and Jacobian regularization (Hoffman et al., 2019) constrain large singular values of Jacobians. Recent efforts further reduce sharpness via SAM (Foret et al., 2020) or project adversarial inputs back to the data manifold using generative defenses (Meng & Chen, 2017; Samangouei et al., 2018). Yet existing methods typically act indirectly, depend on auxiliary models, or require sensitive hyperparameters that hinder deployment. By contrast, DSD operates directly in gradient space, deterministically removing the dominant singular component to suppress the principal curvature contributor while provably preserving descent. Being hyperparameter-free and readily practicable, DSD offers a simple yet principled mechanism for enhancing SNN robustness.

## 6 CONCLUSION AND DISCUSSION

**Conclusion.** In this paper, we experimentally demonstrate that SNNs trained with direct encoding and BPTT can undergo catastrophic model collapse when hetero-training which is common in real-world scenarios. Through theoretical analysis, we show that the repeated inputs of direct encoding combined with gradient accumulation in BPTT induce extremely large spectral radius in the Hessian matrix of the loss function, causing the model parameters to become trapped in precarious local minima. Motivated by these insights, we propose a hyperparameter-free method named Domaint Singular Deflation (DSD): by orthogonally projecting gradients to precisely eliminate their dominant singular components, DSD effectively reduces the Hessian spectral radius. Extensive evaluations under both homogeneous and heterogeneous training conditions demonstrate that DSD substantially enhances SNN robustness, paving the way for safer and more reliable deployments.

**Limitation.** The gradient-based adjustment inherent to DSD induces a deliberate divergence between the gradients actually applied during training and those that would be obtained under an ideal, unmodified regime. While this adjustment markedly bolsters SNNs' robustness, it unavoidably incurs a slight degradation in accuracy when evaluated on unperturbed data. This limitation is also prevalent in existing SOTA methods (Ding et al., 2024b;a; Geng & Li, 2023; Xu et al., 2024) according to Table 1, this calls for more extensive and in-depth future research to overcome.

## ACKNOWLEDGMENT

This work was supported in part by UK Research and Innovation (UKRI) Grant No. EP/Y036786/1, and Horizon Europe Grant No. 101129910.

## REPRODUCIBILITY STATEMENT

The complete code with fixed random seed utilized in this work is publicly available at `https://github.com/Apple26419/SNN_DSD`. All datasets employed in this research, including CIFAR-10, CIFAR-100, Tiny-ImageNet, ImageNet, DVS-CIFAR10, and DVS-Gesture are publicly accessible. Details regarding the hardware, coding environment, and hyperparameter settings used in our experiments are also included in the Appendix. We dedicate to enable researchers to reproduce the results presented in this paper using similar computational setups.

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

# A  ADVERSARIAL ATTACK DETAILS AND CONFIGURATIONS

**FGSM.** FGSM is a simple yet effective technique to generate adversarial examples. In FGSM, given an input $x$ with its true label $y$, a perturbation is computed in the direction of the gradient of the loss function with respect to $x$. The perturbation is defined as:

$$\delta = \epsilon \cdot \text{sign}\left(\nabla_x \mathcal{L}(f(x), y)\right). \tag{16}$$

where $\epsilon$ controls the magnitude of the perturbation and $h(x)$ represents the model's output. The adversarial example is then constructed as $x + \delta$, which is designed to force the model into misclassification.

**PGD.** PGD is an iterative method for generating adversarial examples and can be regarded as an extension of the FGSM. PGD updates the adversarial example iteratively by performing a gradient ascent step and then projecting the result back onto the feasible set defined by the $L_p$-norm constraint. Formally, the update rule is given by:

$$x^{t+1} = \Pi_{\mathcal{B}(x,\epsilon)}\Big(x^t + \alpha \cdot \text{sign}\big(\nabla_x \mathcal{L}(f(x^t), y)\big)\Big). \tag{17}$$

where $\alpha$ denotes the step size, $\mathcal{L}(f(x^t), y)$ is the loss function of the model $h$ with true label $y$, and $\Pi_{\mathcal{B}(x,\epsilon)}$ is the projection operator that projects the perturbed example back into the ball $\mathcal{B}(x, \epsilon) = \{x' : \|x' - x\|_p \le \epsilon\}$. By iterating this process, PGD effectively seeks a perturbation that maximizes the loss while ensuring that the adversarial example remains within the specified perturbation budget.

**APGD.** Auto-PGD (APGD) (Croce & Hein, 2020) is an iterative adversarial attack based on PGD, equipped with an adaptive step-size strategy and a momentum-like update. Given a perturbation budget $\epsilon$, the APGD update is defined as

$$x^{t+1} = \Pi_{\mathcal{B}(x,\epsilon)}\big(x^t + \alpha_t \cdot \text{sign}\big(\nabla_x L(f(x^t), y)\big)\big), \tag{18}$$

where $\alpha_t$ denotes the step size at iteration $t$. APGD further maintains an auxiliary momentum variable:

$$z^t = x^t + \beta_t(x^t - x^{t-1}), \qquad x^{t+1} = \Pi_{\mathcal{B}(x,\epsilon)}\big(z^t + \alpha_t \cdot \text{sign}\big(\nabla_x L(f(z^t), y)\big)\big), \tag{19}$$

where $\beta_t$ is the momentum coefficient.

**Hyperparameters.** For these attack methods, we set $\epsilon = 8/255$ for all experimental cases. For PGD, step size $\alpha = 0.01$ and step number $K = 7$. For APGD, there are two loss versions: APGDCE (Cross Entropy loss) and APGDDLR (Difference of Logits Ratio loss), we use the $L_\infty$-bounded setting with perturbation budget $\epsilon = 8/255$, initial step size $\alpha_0 = 2/255$.

# B  EXPERIMENTAL SETTING

## B.1  SETTINGS FOR PRELIMINARY EXPERIMENTS

In this section, we detail the network architectures and hyperparameter settings used in the preliminary experiments of Sec. 3.1. Any configurations not mentioned here are identical to those in the main experiments (Appendix B.2). For Fig. 2(a), we set the number of epochs to 50, adopt Poisson encoding for rate encoding (Lee et al., 2020), and use a standard convolutional neural networks (CNN) for ANN. All experiments in Figs.2(a–c) employ the hyperparameters listed in Table 6. Direct encoding is performed with $T = 4$, while rate encoding uses $T = 64$.

Table 6: Hyperparameter settings for preliminary experiments.

| Dataset | Optimizer | LeaningRate | WeightDecay | BatchSize |
|---------|-----------|-------------|-------------|-----------|
| CIFAR-10 | SGD | 0.1 | 5e-5 | 128 |

### B.2 SETTINGS FOR MAIN EXPERIMENTS

In our main experiments, all training cases are implemented using PyTorch (Paszke, 2019) with the SpikingJelly (Fang et al., 2023) framework and executed on an NVIDIA GeForce RTX 5090 GPU. For each dataset, we utilize the hyperparameters listed as Table 7, consistently employing the SGD optimizer and setting the membrane time constant $\tau$ to 1.1. We leverage the PyHessian framework (Yao et al., 2020) to compute Hessian eigenvalues[1].

Table 7: Hyperparameter settings for experiments. *(Brock et al., 2021)

| Dataset | Model | LeaningRate | WeightDecay | Epoch | BatchSize | TimeStep |
|---|---|---|---|---|---|---|
| CIFAR-10 | VGG-11 | 0.1 | 5e-5 | 300 | 128 | 4 |
| CIFAR-100 | VGG-11 | 0.1 | 5e-4 | 300 | 128 | 4 |
| TinyImageNet | VGG-16 | 0.1 | 5e-4 | 300 | 128 | 4 |
| ImageNet | NF-ResNet-18* | 0.1 | 1e-5 | 100 | 512 | 4 |
| ImageNet (AT) | ResNet-18 | 0.1 | 1e-5 | 100 | 512 | 4 |
| DVS-CIFAR10 | VGG-11 | 0.05 | 5e-4 | 200 | 128 | 10 |
| DVS-Gesture | VGG-11 | 0.05 | 5e-4 | 200 | 8 | 20 |

## C ANALYSIS OF BOUNDED BPTT FACTORS UNDER LIF DYNAMICS

We make explicit why the constants in the inequalities $\|G_t\| \leq C_G$, $\|D_t\| \leq C_D$, $\left\|\frac{\partial V_{i+1}^{l+1}}{\partial V_i^{l+1}} + \frac{\partial V_{i+1}^{l+1}}{\partial S_i^{l+1}} \frac{\partial S_i^{l+1}}{\partial V_i^{l+1}}\right\| \leq \rho < 1$ exist and are independent of $t$ and $T$.

**(i) Bound on $G_t$.** Write $G_t = \frac{\partial L}{\partial S_t^{l+1}} \frac{\partial S_t^{l+1}}{\partial V_t^{l+1}}$. Let $\phi(\cdot)$ be the surrogate nonlinearity for spikes, with $|\phi'(u)| \leq \kappa$ (e.g., sigmoid with slope $\beta$ gives $\kappa \leq \beta/4$; piecewise-linear surrogates have a fixed maximal slope). For standard losses, $\left\|\frac{\partial L}{\partial z}\right\| \leq C_{\text{loss}}$ uniformly in the logits $z$: for softmax cross-entropy, $H_z = \nabla_z^2 L = \text{Diag}(p) - pp^\top \succeq 0$ gives $\left\|\frac{\partial L}{\partial z}\right\| \leq 1$ and $\|H_z\| \leq \frac{1}{4}$ (binary) or $\leq \frac{1}{2}$ (multi-class). Since $\frac{\partial L}{\partial S_t^{l+1}} = \frac{\partial L}{\partial z_t^{l+1}} \frac{\partial z_t^{l+1}}{\partial S_t^{l+1}}$ and the readout weights are kept bounded by regularization/clipping, $\left\|\frac{\partial z_t^{l+1}}{\partial S_t^{l+1}}\right\| \leq \Lambda_{\text{out}}$. Therefore,

$$\|G_t\| = \left\|\frac{\partial L}{\partial S_t^{l+1}} \frac{\partial S_t^{l+1}}{\partial V_t^{l+1}}\right\| \leq C_{\text{loss}}\Lambda_{\text{out}} \cdot \kappa = C_G, \tag{20}$$

a uniform bound independent of $t, T$.

**(ii) Bound on $D_t$.** For a LIF layer, $V_t^{l+1} = \alpha V_{t-1}^{l+1} + W^{l+1}S_t^l + b$ (plus optional input term) in practical code implementation (Fang et al., 2023), with $\alpha = 1 - \frac{1}{\tau} \in (0,1)$. The local Jacobian w.r.t. $W^l$ at time $t$ is linear in the presynaptic spikes: $D_t = \frac{\partial V_t^{l+1}}{\partial W^l} = \mathcal{L}(S_t^l)$ where $\|S_t^l\| \leq \sqrt{n_{\text{in}}} r_{\text{max}}$ because spikes are binary and $r_{\text{max}} \leq 1$. Hence there exists $C_{\text{in}}$ such that

$$\|D_t\| \leq C_{\text{in}} = C_D. \tag{21}$$

**(iii) Contraction of the recurrent Jacobian chain.** For LIF, $\frac{\partial V_{i+1}^{l+1}}{\partial V_i^{l+1}} = \alpha I$ and $\frac{\partial V_{i+1}^{l+1}}{\partial S_i^{l+1}} = W^{l+1}$, while $\frac{\partial S_i^{l+1}}{\partial V_i^{l+1}} = \phi'(V_i^{l+1})$ with $\|\phi'\|_\infty \leq \kappa$. Therefore

$$\left\|\frac{\partial V_{i+1}^{l+1}}{\partial V_i^{l+1}} + \frac{\partial V_{i+1}^{l+1}}{\partial S_i^{l+1}} \frac{\partial S_i^{l+1}}{\partial V_i^{l+1}}\right\| \leq \alpha + \|W^{l+1}\|\kappa. \tag{22}$$

---

[1]Hyperparameters set as default: $maxIter = 100$, $tol = 1e-3$, where $maxIter$: maximum iterations used to compute each single eigenvalue, $tol$: the relative tolerance between two consecutive eigenvalue computations from power iteration.

Imposing a spectral-norm control $\|W^{l+1}\| \le \Lambda$ with $\alpha + \kappa\Lambda < 1$ yields a uniform contraction rate

$$\rho = \alpha + \kappa\Lambda < 1, \tag{23}$$

so that every time-local Jacobian factor and their products satisfy $\|J_{k:t-1}\| \le \rho^{t-k}$.

**(iv) Bound on the readout Jacobian $B_t$.** Let $z_t$ denote the logits at time $t$ and $B_t = \frac{\partial z_t}{\partial V_t^{l+1}}$. The readout in SNNs is typically linear w.r.t. a hidden state $h_t$ (either the membrane potential $V_t^{l+1}$ or the spike $S_t^{l+1}$):

$$z_t = W_{\text{out}} h_t + b, \qquad h_t \in \{ V_t^{l+1},\ S_t^{l+1} = \phi(V_t^{l+1}) \}, \tag{24}$$

and $z = \frac{1}{\alpha_T} \sum_{t=1}^T z_t$ with $\alpha_T \ge 1$ (e.g., $\alpha_T = T$ for averaging). Let $\Lambda_{\text{out}} = \|W_{\text{out}}\|$ (controlled by weight decay / clipping / spectral normalization), and let $\kappa := \sup_u |\phi'(u)|$ be the maximal slope of the surrogate nonlinearity $\phi$.

- **Case 1 (direct-$V$ readout).** If $h_t = V_t^{l+1}$, then $B_t = \frac{\partial z_t}{\partial V_t^{l+1}} = W_{\text{out}}$, hence

$$\|B_t\| \le \|W_{\text{out}}\| = \Lambda_{\text{out}}. \tag{25}$$

- **Case 2 (spike readout).** If $h_t = S_t^{l+1} = \phi(V_t^{l+1})$, then $B_t = \frac{\partial z_t}{\partial S_t^{l+1}} \frac{\partial S_t^{l+1}}{\partial V_t^{l+1}} = W_{\text{out}} \phi'(V_t^{l+1})$, hence

$$\|B_t\| \le \|W_{\text{out}}\| \, \|\phi'(V_t^{l+1})\| \le \Lambda_{\text{out}} \kappa. \tag{26}$$

Combining the cases, there exists a uniform constant

$$C_B = \kappa \, \Lambda_{\text{out}} \tag{27}$$

such that $\|B_t\| \le C_B$ for all $t$, independent of $T$. Consequently, $\|H_t\| = \|B_t^\top H_{z,t} B_t\| \le C_B^2 \|H_{z,t}\| \le C_B^2 C_z$.

**(v) Bound on $H_{z,t}$.** For softmax cross-entropy, $H_{z,t} = \text{Diag}(p_t) - p_t p_t^\top$ has $\|H_{z,t}\| \le \frac{1}{2}$ (and $\le \frac{1}{4}$ for binary); for squared loss, $\|H_{z,t}\| \le 1$. Hence there exists a global $C_z < \infty$ with $\|H_{z,t}\| \le C_z$ for all $t$.

**(vi) Bound on $J_t^W$.** We now show why the bound $\|J_t^W\| \le \frac{C_D}{1-\rho} = C_J$ holds. Recall that $\|D_k\| \le C_D$ for all $k$, and $\|J_{k:t-1}\| \le \rho^{t-k}$ with $\rho \in (0,1)$. Using the triangle inequality and the sub-multiplicativity of the operator norm, we obtain

$$
\begin{aligned}
\|J_t^W\| &= \left\| D_t + \sum_{k<t} J_{k:t-1} D_k \right\| \\
&\le \|D_t\| + \sum_{k<t} \|J_{k:t-1} D_k\| \\
&\le C_D + \sum_{k<t} \|J_{k:t-1}\| \, \|D_k\| \\
&\le C_D + \sum_{k<t} \rho^{t-k} C_D \\
&= C_D \left(1 + \sum_{q=1}^{t-1} \rho^q\right) \\
&\le C_D \sum_{q=0}^{\infty} \rho^q \\
&= \frac{C_D}{1-\rho} = C_J.
\end{aligned}
\tag{28}
$$

Here we re-indexed with $q = t - k$ and used the geometric series bound $\sum_{q=0}^{\infty} \rho^q = \frac{1}{1-\rho}$. Thus $\|J_t^W\|$ is uniformly bounded by $C_J$, independent of $t$ and $T$.

# D DATASET

**CIFAR-10.** The CIFAR-10 dataset (Krizhevsky et al., 2009) consists of 60,000 color images, each of size 32×32 pixels, divided into 10 different classes, such as airplanes, cars, birds, cats, and dogs. Each class has 6,000 images, with 50,000 images used for training and 10,000 for testing. Normalization, random horizontal flipping, random cropping with 4 padding, and CutOut (DeVries & Taylor, 2017) are applied for data augmentation.

**CIFAR-100.** The CIFAR-100 dataset (Krizhevsky et al., 2009) consists of 60,000 color images, each of size 32×32 pixels, categorized into 100 different classes. Each class contains 600 images, with 500 used for training and 100 for testing. The same processing methods as for dataset CIFAR-10 are applied to dataset CIFAR-100.

**Tiny-ImageNet.** The Tiny-ImageNet dataset is a scaled-down version of the ImageNet dataset (Deng et al., 2009). It contains 200 different classes, with 500 training images and 50 testing images per class, resulting in a total of 100,000 training images and 10,000 testing images. Each image is resized to 64×64 pixels. Normalization, random horizontal flipping, and random cropping with 4 padding are applied for data augmentation for the Tiny-ImageNet dataset.

**ImageNet.** We evaluate on the ILSVRC-2012 ImageNet dataset (Deng et al., 2009), which contains ∼1.28M training images and 50,000 validation images spanning 1,000 classes. Images are of variable resolution; following common practice and our implementation, training augmentation includes RandomResizedCrop to $224 \times 224$, RandomHorizontalFlip, conversion to tensors, and channel-wise normalization. For test, images are resized to have a shorter side of 256 pixels and then center-cropped to $224 \times 224$ before applying the same normalization.

**DVS-CIFAR10.** The DVS-CIFAR-10 dataset (Li et al., 2017) is a neuromorphic version of the traditional CIFAR-10 dataset. DVS-CIFAR10 captures the visual information using a Dynamic Vision Sensor (DVS), which records changes in the scene as a series of asynchronous events rather than as a sequence of frames. The dataset consists of recordings of 10 object classes, corresponding to the original CIFAR-10 categories, with each object presented in front of a DVS camera under various conditions. The dataset contains 10,000 128×128 images, of which 9,000 are used as the training set and the remaining 1,000 as the test set.

**DVS-Gesture.** The DVS-Gesture dataset (Amir et al., 2017) is a neuromorphic dataset, consisting of 11 different hand gesture classes, such as hand clapping, arm rolling, and air guitar, performed by 29 subjects under various lighting conditions. Each gesture is represented by a sequence of events rather than frames. The dataset contains 1,176 training samples and 288 testing samples.

# E EVALUATION OF ADDITIONAL TRAINING COMPUTATIONAL OVERHEAD

During training, we benchmarked its SVD overhead against a vanilla SNN using a VGG-11 framework. Our measurements, as Table 8, show **no increase in memory usage**, and on an NVIDIA GeForce RTX 4070 Ti, DSD adds only **around 0.1s of extra training time per batch**, this is nearly neglectable. In summary, although DSD does introduce slight training overhead, the increase is minimal.

Table 8: Computational overhead evaluation.

| Dataset | BatchSize | BatchNum | DSD | Memory | AvgTime perEpoch | AvgTime perBatch |
|---|---|---|---|---|---|---|
| CIFAR-10, 100 | 128 | 390 | ✗ | 1.4102GB | 34s | 0.0872s |
| | | | ✓ | 1.4102GB | 63s | 0.1616s ▲0.0743 |
| DVS-CIFAR10 | 128 | 71 | ✗ | 5.1797GB | 38s | 0.5352s |
| | | | ✓ | 5.1797GB | 46s | 0.6479s ▲0.1127 |
| DVS-Gesture | 8 | 73 | ✗ | 8.4727GB | 63s | 0.8630s |
| | | | ✓ | 8.4727GB | 70s | 0.9589s ▲0.0959 |

Table 9: Performance of DSD with different attack methods (%). This is detailed experimental results for Fig. 4.

| Attack | $\epsilon = 0$ | 2 | 4 | 6 | 8 | 16 | 32 | 64 | 128 |
|---|---|---|---|---|---|---|---|---|---|
| **CIFAR-10** | | | | | | | | | |
| SNN FGSM WB | 93.75 | 17.06 | 14.22 | 11.78 | 8.19 | 4.21 | 1.99 | 0.58 | 0.00 |
| SNN FGSM BB | 93.75 | 24.13 | 18.41 | 13.86 | 10.26 | 7.88 | 3.82 | 1.48 | 0.59 |
| SNN PGD WB | 93.75 | 2.37 | 1.01 | 0.34 | 0.03 | 0.00 | 0.00 | 0.00 | 0.00 |
| SNN PGD BB | 93.75 | 4.01 | 2.80 | 1.20 | 0.89 | 0.02 | 0.00 | 0.00 | 0.00 |
| DSD FGSM WB | 90.21 | 59.75 | 58.22 | 57.00 | 55.86 | 50.36 | 39.82 | 26.61 | 8.01 |
| DSD FGSM BB | 90.21 | 81.42 | 79.37 | 76.65 | 74.90 | 67.18 | 49.30 | 31.09 | 12.81 |
| DSD PGD WB | 90.21 | 40.78 | 37.46 | 34.50 | 31.44 | 22.04 | 6.36 | 0.03 | 0.00 |
| DSD PGD BB | 90.21 | 57.88 | 54.09 | 47.80 | 43.15 | 30.73 | 12.11 | 0.91 | 0.24 |
| **CIFAR-100** | | | | | | | | | |
| SNN FGSM WB | 72.39 | 9.37 | 7.42 | 5.46 | 4.55 | 2.35 | 1.19 | 0.33 | 0.00 |
| SNN FGSM BB | 72.39 | 13.26 | 12.11 | 10.84 | 9.16 | 5.31 | 2.07 | 1.47 | 0.49 |
| SNN PGD WB | 72.39 | 2.53 | 1.15 | 0.50 | 0.19 | 0.02 | 0.00 | 0.00 | 0.00 |
| SNN PGD BB | 72.39 | 3.65 | 2.68 | 1.89 | 0.78 | 0.15 | 0.02 | 0.00 | 0.00 |
| DSD FGSM WB | 70.26 | 27.89 | 26.65 | 24.83 | 23.81 | 14.81 | 8.38 | 3.36 | 0.20 |
| DSD FGSM BB | 70.26 | 40.89 | 38.67 | 35.67 | 31.55 | 21.63 | 13.31 | 9.03 | 1.01 |
| DSD PGD WB | 70.26 | 20.52 | 15.78 | 12.35 | 8.09 | 2.37 | 0.00 | 0.00 | 0.00 |
| DSD PGD BB | 70.26 | 27.99 | 26.01 | 23.84 | 18.37 | 9.01 | 1.07 | 0.00 | 0.00 |
| **TinyImageNet** | | | | | | | | | |
| SNN FGSM WB | 56.82 | 9.42 | 7.60 | 4.82 | 3.51 | 1.53 | 0.67 | 0.00 | 0.00 |
| SNN FGSM BB | 56.82 | 16.27 | 15.59 | 13.69 | 12.46 | 8.20 | 4.80 | 2.74 | 1.28 |
| SNN PGD WB | 56.82 | 2.98 | 1.46 | 0.89 | 0.14 | 0.00 | 0.00 | 0.00 | 0.00 |
| SNN PGD BB | 56.82 | 4.07 | 2.99 | 2.12 | 1.67 | 0.99 | 0.46 | 0.00 | 0.00 |
| DSD FGSM WB | 54.54 | 22.80 | 21.74 | 21.00 | 19.50 | 12.66 | 6.58 | 1.58 | 0.02 |
| DSD FGSM BB | 54.54 | 34.63 | 31.02 | 29.40 | 27.68 | 20.22 | 10.35 | 3.20 | 1.38 |
| DSD PGD WB | 54.54 | 18.42 | 15.70 | 13.22 | 12.02 | 5.78 | 2.96 | 0.20 | 0.00 |
| DSD PGD BB | 54.54 | 24.92 | 20.66 | 16.82 | 14.84 | 6.55 | 4.47 | 2.10 | 0.25 |
| **ImageNet** | | | | | | | | | |
| SNN FGSM WB | 57.84 | 10.75 | 8.59 | 6.73 | 4.99 | 1.56 | 0.14 | 0.01 | 0.00 |
| SNN FGSM BB | 57.84 | 12.13 | 11.43 | 10.25 | 8.46 | 5.42 | 3.93 | 1.25 | 0.35 |
| SNN PGD WB | 57.84 | 1.02 | 0.48 | 0.13 | 0.01 | 0.00 | 0.00 | 0.00 | 0.00 |
| SNN PGD BB | 57.84 | 9.19 | 6.05 | 4.06 | 3.67 | 2.02 | 1.79 | 0.05 | 0.00 |
| DSD FGSM WB | 53.47 | 18.62 | 17.31 | 15.65 | 14.69 | 10.98 | 5.30 | 2.01 | 0.02 |
| DSD FGSM BB | 53.47 | 27.68 | 25.58 | 24.44 | 22.40 | 16.24 | 8.96 | 4.90 | 2.16 |
| DSD PGD WB | 53.47 | 11.63 | 8.12 | 6.33 | 4.30 | 2.97 | 1.50 | 0.00 | 0.00 |
| DSD PGD BB | 53.47 | 16.10 | 12.22 | 8.33 | 5.68 | 4.09 | 1.67 | 0.20 | 0.04 |

# F    DETAILED EXPERIMENTAL RESULT

In this section, we present the full experimental results underlying Figs. 4, 5, 6, and 7 as Tables 9, 11, and 12, respectively, from the main text.

Table 10: Comparison of hetero-training performance degradation (%). This is detailed experimental results for Fig. 5, where "*" indicates self-implementation.

| Methods | $b$ | CIFAR-10, c/+p_0 | | | CIFAR-10, p/+c_0 | | | CIFAR-100, c/+p_0 | | | CIFAR-100, p/+c_0 | | |
|---|---|---|---|---|---|---|---|---|---|---|---|---|---|
| | | Clean | FGSM | PGD | Clean | FGSM | PGD | Clean | FGSM | PGD | Clean | FGSM | PGD |
| SNN | 0* | 93.75 | 8.19 | 0.03 | 91.16 | 38.20 | 14.07 | 72.39 | 4.55 | 0.19 | 69.69 | 16.31 | 8.49 |
| | 1* | 12.97 | 0.34 | 0.01 | 18.54 | 4.07 | 1.98 | 3.75 | 0.07 | 0.01 | 3.41 | 1.40 | 0.20 |
| RAT (Ding et al., 2022) | 0 | 93.01* | 12.63* | 0.05* | 90.74 | 45.23 | 21.16 | 71.00* | 5.77* | 0.15* | 70.89 | 25.86 | 10.38 |
| | 1* | 27.24 | 8.76 | 0.03 | 29.01 | 27.67 | 8.00 | 19.20 | 2.59 | 0.04 | 20.11 | 10.78 | 3.81 |
| DLIF (Ding et al., 2024a) | 0 | 92.22 | 13.24 | 0.09 | 88.91 | 56.71 | 40.30 | 70.79 | 6.95 | 0.08 | 66.33 | 36.83 | 24.25 |
| | 1* | 18.40 | 3.50 | 0.02 | 14.75 | 19.02 | 5.74 | 9.98 | 1.02 | 0.02 | 11.40 | 8.02 | 2.97 |
| FEEL (Xu et al., 2024) | 0 | 93.29 | 44.96 | 28.35 | 90.20* | 59.08* | 39.87* | 73.79 | 9.60 | 2.04 | 69.79 | 18.07 | 11.07 |
| | 1* | 20.01 | 8.19 | 2.77 | 13.99 | 17.50 | 7.07 | 12.84 | 2.00 | 0.10 | 13.86 | 9.39 | 1.89 |
| DSD | 0* | 90.21 | 55.86 | 31.44 | 86.62 | 74.43 | 44.38 | 70.26 | 23.81 | 9.09 | 64.21 | 43.91 | 27.11 |
| | 1* | 40.58 | 33.70 | 22.87 | 38.61 | 40.09 | 20.77 | 27.40 | 12.54 | 3.97 | 22.08 | 19.16 | 13.71 |

Table 11: DSD performance in hetero-training (%). The data in parentheses represents the difference from the baseline ($b = 0$). This is detailed experimental results for Fig. 6.

| $b$ | Hetero-training: c/+p_0 | | | | Hetero-training: p/+c_0 | | | |
|---|---|---|---|---|---|---|---|---|
| | CIFAR-10 | CIFAR-100 | TinyImageNet | ImageNet | CIFAR-10 | CIFAR-100 | TinyImageNet | ImageNet |
| **Clean Inference** | | | | | | | | |
| 0 | 90.21 | 70.26 | 54.54 | 53.47 | 86.62 | 64.21 | 46.30 | 49.78 |
| 1 | 40.58 (-49.63) | 27.40 (-42.86) | 21.45 (-33.09) | 23.54 (-29.93) | 38.61 (-48.01) | 22.08 (-42.13) | 19.30 (-27.00) | 22.69 (-27.09) |
| 2 | 31.08 (-59.13) | 12.76 (-57.50) | 9.64 (-44.90) | 10.53 (-42.94) | 29.95 (-56.67) | 11.58 (-52.63) | 8.95 (-37.35) | 9.03 (-40.75) |
| 5 | 12.07 (-78.14) | 4.18 (-66.08) | 3.98 (-50.56) | 5.67 (-47.80) | 12.02 (-74.60) | 3.77 (-60.44) | 2.72 (-43.58) | 3.44 (-46.34) |
| **FGSM Inference** | | | | | | | | |
| 0 | 55.86 | 23.81 | 19.50 | 14.59 | 74.43 | 43.91 | 30.87 | 26.83 |
| 1 | 33.70 (-22.16) | 12.54 (-11.27) | 14.67 (-4.83) | 11.45 (-3.14) | 40.09 (-34.34) | 19.16 (-24.75) | 12.04 (-18.83) | 11.68 (-15.15) |
| 2 | 33.29 (-22.57) | 13.70 (-10.11) | 14.78 (-4.72) | 11.12 (-3.47) | 38.22 (-36.21) | 17.31 (-26.60) | 10.11 (-20.76) | 8.64 (-18.19) |
| 5 | 34.85 (-21.01) | 14.93 (-8.88) | 15.56 (-3.94) | 11.86 (-2.73) | 31.58 (-42.85) | 8.49 (-35.42) | 5.80 (-25.07) | 3.10 (-23.73) |
| **PGD Inference** | | | | | | | | |
| 0 | 31.44 | 9.09 | 12.02 | 4.30 | 44.38 | 27.11 | 18.21 | 9.12 |
| 1 | 22.87 (-8.57) | 3.97 (-5.12) | 8.89 (-3.13) | 2.67 (-1.63) | 20.77 (-23.61) | 13.71 (-13.40) | 12.44 (-5.77) | 4.90 (-4.22) |
| 2 | 21.23 (-10.21) | 3.34 (-5.75) | 8.67 (-3.35) | 2.21 (-2.09) | 18.72 (-25.66) | 11.22 (-15.89) | 11.90 (-6.31) | 3.38 (-5.74) |
| 5 | 21.69 (-9.75) | 3.01 (-6.08) | 9.02 (-3.00) | 1.88 (-2.42) | 14.31 (-30.07) | 7.93 (-19.18) | 8.78 (-9.43) | 1.94 (-7.18) |

Table 12: Performance comparison with different PGD step number on CIFAR-10 (%). This is detailed experimental results for Fig. 7.

| Method | $K = 7$ | 10 | 15 | 20 | 30 | 40 | 50 | 60 | 70 | 80 |
|---|---|---|---|---|---|---|---|---|---|---|
| SNN | 0.03 | 0.01 | 0.00 | 0.00 | 0.00 | 0.00 | 0.00 | 0.00 | 0.00 | 0.00 |
| SNN (AT) | 14.07 | 13.48 | 12.99 | 12.03 | 11.36 | 10.79 | 10.49 | 10.44 | 10.42 | 10.41 |
| DSD | 31.44 | 30.34 | 29.12 | 27.66 | 26.40 | 25.74 | 25.50 | 25.46 | 25.45 | 25.44 |
| DSD (AT) | 44.38 | 43.10 | 42.56 | 41.29 | 40.78 | 40.01 | 39.80 | 39.78 | 39.77 | 39.77 |

## G    STATEMENT OF LARGE LANGUAGE MODEL (LLM) USAGE

In the preparation of this manuscript, an LLM was employed to assist with non-scientific tasks. These included polishing the English writing for clarity and style, providing suggestions for figure design and color schemes, supporting LaTeX formatting and typesetting, and drafting this statement.

