# OpenReview forum: "Robustify Spiking Neural Networks via Dominant Singular Deflation under Heterogeneous Training Vulnerability"
_ICLR.cc/2026/Conference — ICLR 2026 Poster_

### Official Review · Reviewer_RTMm · 2025-10-16

**Soundness:** 3
**Presentation:** 3
**Contribution:** 3
**Rating:** 6
**Confidence:** 4

**Summary:**

1. Research Question
- The paper addresses the instability and vulnerability of Spiking Neural Networks (SNNs) under heterogeneous training conditions and adversarial perturbations.
- Key problem: Why do SNNs collapse under small distribution shifts, and can this be mitigated by stabilizing their spectral dynamics during training?

2. Method Proposed: Dominant Singular Deflation (DSD)
- DSD is a spectral regularization technique applied during backpropagation.
- Instead of modifying weights or data, DSD removes the dominant singular component from the gradient matrix (Δ W) at each update step.
- DSD suppresses the gradient’s most unstable direction, aiming to reduce spectral explosion and improve training stability and robustness without introducing new hyperparameters or architectural changes.

3. Theoretical Part
- The authors conduct theoretical analysis based on the spectral properties of the Jacobian and Hessian during BPTT.
- The paper formalizes *heterogeneous training vulnerability* as a consequence of the exponential growth of the Hessian’s spectral radius over time steps.
- It shows (under simplifying assumptions) that the dominant singular direction of (Δ W) aligns across time, amplifying instability.
- Removing the dominant singular direction theoretically reduces the Hessian’s largest eigenvalue, ensuring descent and smoother loss curvature.

4. Experimental Part
- Experiments on CIFAR-10/100, TinyImageNet, and event-based datasets show: DSD stabilizes training (smoother loss, bounded gradient norms).
- DSD Improves robustness under FGSM, PGD without adversarial training, maintains or slightly improves clean accuracy.
- Spectral diagnostics (Hessian radius, singular value histograms) support the theoretical mechanism.

**Strengths:**

This work reminds me of a relevant paper: SNN-RAT (https://openreview.net/forum?id=xwBdjfKt7_W). The two works do have very similar views—both aim to improve robustness by controlling the spectral properties of SNNs (singular values/Lipschitz constants). However, they have subtle but important differences in their conceptual approach, theoretical approach, and technical implementation. Therefore, I will use SNN-RAT as a comparison to discuss this paper.


1. Theoretical Part

- Presents a novel *optimization-space* spectral stabilization method, contrasting prior *parameter-space* approaches such as SNN-RAT, which constrain the largest singular value of (W). DSD instead regularizes the gradient matrix Δ W, introducing a fresh angle on robustness grounded in training dynamics.
- Provides a clear and mathematically coherent link between Hessian spectral growth, gradient alignment, and heterogeneous training instability — a valuable conceptual contribution to understanding SNN optimization behavior.
- Demonstrates theoretical descent guarantees via spectral deflation, suggesting that DSD maintains convergence while suppressing dominant curvature directions.
- DSO is elegant, minimal, and free of additional hyperparameters, making it an analytically interpretable robustness mechanism.

2. Experimental Part

- Empirical results show that DSD improves both training stability and adversarial robustness without adversarial training, which is a notable distinction from previous works relying on adversarial data augmentation like SNN-RAT/HoSNN.
- Consistent improvements across CIFAR-10/100, TinyImageNet, and DVS datasets, while preserving or slightly improving clean accuracy.
- Spectral diagnostics (gradient singular spectra, Hessian radius) align closely with the theory, strengthening internal validity.
- The evaluation includes white-box, black-box scenarios, reducing concerns about gradient obfuscation.

3. Method Scalability

- The DSD operation is simple and parameter-free, requiring only rank-1 spectral deflation per layer, theoretically lightweight compared to full spectral regularization in SNN-RAT.
- The method integrates seamlessly into standard backpropagation and does not alter network architecture, making it easy to adopt in existing SNN frameworks.
- DSO can reduce the reliance on adversarial samples in SNN adversarial training, which greatly reduces the amount of computation while providing adversarial robustness.

**Weaknesses:**

1. Theoretical Part
Overall, the paper's proof follows the path of "spectral analysis of the Jacobian product → increasing the radius of the Hessian spectrum → shrinking the Deflation spectrum → improving stability." This is intuitively reasonable.

- It would be helpful for the rigor and clarity of the paper if the author could clearly state the key assumptions in each theorem. In particular, it would deepen our understanding of the problem by indicating when the core theorems proposed fail.

2. Experimental Part
Overall, I think the experimental part is clear, comprehensive, and rigorous.
- More attack methods like Gaussion noise, APGD (https://arxiv.org/pdf/2003.01690) could be tested. Only FGSM and PGD are tested in the current paper.


3. Method Scalability
- The proposed DSD step requires layer-wise SVD or power iteration, which is computationally demanding for large-scale or convolutional SNNs; the paper should quantify this cost in the main paper.
- DSD depends on full gradient access and is thus incompatible with neuromorphic or local learning implementations, limiting its deployability on real spiking hardware.
- The direct modification of the principal components of the gradient is a concern. Even though the paper provides good results overall, unpredictable results may occur when interacting with other adaptive optimizer such as Adam and RMSProp.

**Questions:**

1.Could the authors explicitly state the key assumptions/conditions in each theorem?   For example:
  * Under what conditions do the contraction and boundedness assumptions on the Jacobian fail?
  * Are there cases where the spectral radius of the Hessian would not scale linearly with time steps (T)?
  * What happens if the “approximate time-invariance” assumption of the Jacobian is violated (e.g., due to noise, dropout, or adaptive thresholds)?

2. Could the authors consider evaluating the robustness under additional perturbations such as Gaussian noise or stronger adversarial attacks e.g., APGD?

3. Could the authors quantify DSD's computational cost (e.g., time per iteration, FLOPs, or scaling with network size)

4. Could the authors discuss potential adaptations or approximations that make DSD compatible with neuromorphic chips?

5.  Have the authors observed any instabilities or performance inconsistencies during DSD training process or when DSO is combined with different optimizers?

---

> ### Author Response · Authors · 2025-11-22
> **Response to Reviewer RTMm, part 1/4**
>
> ### **Question 1**
>
> Could the authors explicitly state the key assumptions/conditions in each theorem?
>
> ### **Response**
>
> Thanks for your interest. We respond to these questions as follows.
>
> ---
>
> ### **1.1 Under what conditions do the contraction and boundedness assumptions on the Jacobian fail?**
>
> ### **Response 1.1**
>
> In our theoretical analysis, the contraction and boundedness assumptions on the Jacobian follow directly from two basic modeling choices:
> (i) the use of standard LIF neurons with a built-in membrane-leak mechanism, and
> (ii) the use of bounded surrogate gradients.
>
> Under these conditions, each single-step LIF update is contractive, and the surrogate gradient remains uniformly bounded, ensuring that the norm of each Jacobian block is controlled by a constant $C_J$.
>
> These assumptions may fail in a few extreme or deliberately constructed cases:
>
> 1. **Neuron dynamics become non-contractive.**
>    When the leakage parameter is set extremely close to $1$ (or equivalently, when the residual weight on $V_{t-1}$ approaches or exceeds $1$), the membrane potential no longer decays and the system becomes non-contractive.
>    A typical example is the IF neuron, which does not satisfy the contraction property due to the absence of leakage.
>
> 2. **Artificially introduced strong positive feedback.**
>    If very large recurrent gains are imposed so that the mapping behaves as $V_t \approx \alpha V_{t-1}$ with $\alpha \ge 1$, the Jacobian ceases to be contractive.
>    In practical SNN architectures, however, normalization-based mechanisms (e.g., BN/TEBN) prevent such behavior, making this case negligible under standard training.
>
> 3. **Extremely steep surrogate gradients.**
>    If the surrogate gradient has an abnormally large slope near the firing threshold, $\partial S_t / \partial V_t$ can become excessively large, destroying any uniform upper bound on the Jacobian.
>
> Overall, these situations correspond to intentionally constructed or highly atypical parameter choices. In standard SNN practice—using conventional LIF neurons and commonly adopted surrogate gradients—the contraction and boundedness assumptions hold robustly, and such failure cases essentially do not occur.
>
> ### **1.2 Are there cases where the spectral radius of the Hessian would not scale linearly with time steps (T)?**
>
> ### **Response 1.2**
>
> Theorem 1 and Theorem 2 state that, under our assumptions, the spectral radius of the Hessian grows at most $O(T)$, and under direct encoding together with approximate time-invariance of the Jacobian, this upper bound becomes tight and yields a linear growth rate $\Theta(T)$.
>
> This linear dependence holds only when the corresponding assumptions are satisfied. Linear scaling therefore represents the typical or worst-case behavior under those conditions, but it is not the only possible trend.
>
> #### **(a) Cases where the scaling can be sub-linear**
>
> The spectral radius may grow more slowly than linearly when certain mechanisms suppress long-range temporal contributions. Examples include:
>
> * strong regularization or explicit spectral constraints (e.g., Jacobian/Hessian norm penalties), which reduce the per-step curvature contribution
>
> In such cases, the theoretical upper bound $O(T)$ still holds, but the observed growth can be sub-linear or may even approach a plateau.
>
> #### **(b) Cases where the growth can exceed linear**
>
> If the contraction or boundedness assumptions from Question 1.1 are violated, for example:
>
> * the Jacobian exhibits temporal amplification rather than decay
> * certain parts of the system repeatedly encounter strong positive feedback
>
> then the $O(T)$ bound no longer applies. The spectral radius of the Hessian may grow super-linearly or even approach exponential behavior. This no longer corresponds to the “linear accumulation through time” typical of BPTT, but instead reflects an underlying dynamical instability of the system.

---

> ### Author Response · Authors · 2025-11-22
> **Response to Reviewer RTMm, part 2/4**
>
> ### **1.3 What happens if the “approximate time-invariance” assumption of the Jacobian is violated (e.g., due to noise, dropout, or adaptive thresholds)?**
>
> ### **Response 1.3**
>
> When noise, dropout, adaptive thresholds, or other sources of temporal variation are introduced, the Jacobians across time steps become more heterogeneous, and the principal singular direction is no longer consistent across time. As a result, the rank-1 structure becomes weaker, and the strong alignment effect predicted by Theorem 2 is reduced.
>
> Meanwhile, Hessian growth may become irregular. Although the $O(T)$ upper bound from Theorem 1 still holds, the empirical behavior may change:
>
> 1. the growth rate can become sub-linear
> 2. the curvature may exhibit phase-dependent fluctuations rather than following a unified trend
>
> In such cases, DSD remains effective but operates in a more local manner. Even without time-invariance, whenever certain training phases produce a noticeably dominant curvature direction, the Hessian still becomes locally concentrated along that direction.
>
> DSD continues to suppress the most prominent curvature direction at each step, effectively reducing the local spectral radius. The difference is that its action becomes local in time, rather than relying on the globally rank-1-dominated structure assumed by Theorem 2.
>
> ---
>
> ### **Question 2**
>
> Could the authors consider evaluating the robustness under additional perturbations such as Gaussian noise or stronger adversarial attacks (e.g., APGD)?
>
> ### **Response**
>
> We thank the reviewer for the constructive suggestion. To maintain consistency and comparability with prior SNN robustness studies, our main experiments focus on the two most widely adopted adversarial attacks—FGSM and PGD—which constitute the standard evaluation protocol used by existing SOTA methods. This ensures that our results remain directly aligned with prevailing benchmarks in the field.
>
> At the same time, we fully agree with the reviewer’s interest in stronger adversarial evaluations. In response, we have added a new subsection under homogeneous training (Sec. 4.1) reporting robustness under APGD. The newly added content is highlighted in blue in the revised manuscript, and specific results are listed as follows. The results show that DSD consistently improves robustness under APGD, following the same trend observed under FGSM and PGD.
>
> > *Stronger white box attack. Beyond conventional white-box attacks, we further evaluate whether DSD can defend against stronger adversaries. We employ the APGD attack, which incorporates adaptive step-size control and a momentum-like update during iterations, enabling more efficient exploration of the loss landscape and avoiding local optima to generate stronger adversarial samples. Two variants of APGD are used—APGDCE and APGDDLR—with detailed descriptions and hyperparameter settings provided in Appendix A. As shown in the Table, our method achieves the highest accuracy under all attack settings except for APGDDLR on CIFAR-100 with AT, where it slightly lags behind DLIF. These results demonstrate that DSD remains effective and superior even against stronger white box attacks.*
>
> **Vanilla training**
>
> | Methods     | CIFAR10 | | CIFAR-100 | |
> |-------------|----------------|------------------|-------------------|--------------------|
> | | APGDCE | APGDDLR | APGDCE | APGDDLR |
> | SNN         | 0.61           | 2.06             | 0.09              | 0.12               |
> | DLIF        | 0.05           | 0.03             | 0.02              | 0.18               |
> | HoSNN       | 10.35          | 27.39            | 2.55              | 0.02               |
> | DSD (Ours)  | **22.19**          | **29.77**            | **5.99**              | **5.98**               |
>
> **Adversarial training**
>
> | Methods     | CIFAR10 | | CIFAR-100 | |
> |-------------|----------------|------------------|-------------------|--------------------|
> | | APGDCE | APGDDLR | APGDCE | APGDDLR |
> | SNN         | 10.98          | 17.72            | 5.87              | 6.20               |
> | DLIF        | 35.09          | 39.85            | 20.68             | **24.21**              |
> | HoSNN       | 38.89          | 37.94            | 12.55             | 13.66              |
> | DSD (Ours)  | **47.08**          | **44.62**            | **22.86**             | 23.04              |

---

> ### Author Response · Authors · 2025-11-22
> **Response to Reviewer RTMm, part 3/4**
>
> ### **Question 3**
>
> Could the authors quantify DSD's computational cost (e.g., time per iteration, FLOPs, or scaling with network size)?
>
> ### **Response**
>
> Thank you for the suggestion. We agree that quantifying the computational overhead of DSD is important for understanding its practicality. We have explicitly summarized the computational cost of DSD in Appendix E, where we report the per-epoch and per-batch running time, memory consumption, and the number of additional operations incurred by DSD.
>
> As shown in Table 8 (Appendix E), DSD:
>
> * does not introduce any additional memory overhead
> * increases the per-batch computation time by only 0.07–0.11 seconds
>
> This quantitative evaluation reflects the actual computational footprint of running DSD in standard SNN training pipelines.
>
> ---
>
> ### **Question 4**
>
> Could the authors discuss potential adaptations or approximations that make DSD compatible with neuromorphic chips?
>
> ### **Response**
>
> We appreciate the reviewer’s interest in the deployment feasibility of DSD on neuromorphic chips. We clarify that DSD is used exclusively during the training phase. The trained SNN model (its weights and neuron parameters) is then deployed to neuromorphic hardware.
>
> Because neuromorphic chips are involved only during inference, and DSD does not alter the inference-time computation graph nor introduce any additional operations during deployment, DSD does not need to run on the chip and therefore requires no hardware-side adaptation.
>
> Nonetheless, exploring whether a lightweight approximation of DSD could be incorporated into on-chip training frameworks (such as online learning or local training rules) could be an interesting direction for future work.

---

> ### Author Response · Authors · 2025-11-22
> **Response to Reviewer RTMm, part 4/4**
>
> ### **Question 5**
>
> Have the authors observed any instabilities or performance inconsistencies during DSD training or when DSD is combined with different optimizers?
>
>
> ### **Response**
>
> We appreciate the reviewer’s question regarding potential instabilities during DSD training, and we did encounter such situations. To answer clearly, we first provide the exact SVD decomposition mechanism used in our code (included in the supplementary material, path: \DSD code\utils\tvc.py):
>
> ```
> try:
>     U, S, Vh = torch.linalg.svd(grad_mat, full_matrices=False)
> except RuntimeError:
>     try:
>         U, S, Vh = torch.linalg.svd(
>             grad_mat + 1e-6 * torch.ones_like(grad_mat),
>             full_matrices=False,
>             driver="gesvd"
>         )
>     except RuntimeError:
>         return grad
> ```
>
> ---
>
> ### **What instability did we observe?**
>
> When using SGD (the optimizer adopted in most prior SNN robustness works), we occasionally encountered the following PyTorch SVD error when using:
>
> ```
> U, S, Vh = torch.linalg.svd(grad_mat, full_matrices=False)
> ```
>
> The error message is:
>
> ```
> UserWarning: torch.linalg.svd: During SVD computation with the selected cusolver driver, batches 0 failed to converge. A more accurate method will be used to compute the SVD as a fallback.
> ```
>
> This occurs with extremely low probability—roughly once every 200 epochs, corresponding to fewer than 1 out of 78,000 batches on CIFAR-10/100 with batch size 128. When this happens, the update for that batch is skipped.
>
> Even if we ignore the warning and simply skip the batch, the overall training performance remains unaffected, because dropping 1 batch out of tens of thousands has no measurable impact on optimization.
>
> ---
>
> ### **How did we resolve it?**
>
> To ensure robustness of code running and reproducibility, we include a fallback step:
>
> ```
> U, S, Vh = torch.linalg.svd(
>     grad_mat + 1e-6 * torch.ones_like(grad_mat),
>     full_matrices=False,
>     driver="gesvd"
> )
> ```
>
> This introduces:
>
> * a tiny perturbation (1e-6) to avoid exact singularity
> * a switch to a more stable SVD driver (gesvd), which is slower but more numerically robust
>
> This guarantees that the SVD always converges and that training proceeds smoothly without skipping batches.
>
> ---
>
> ### **Why does it happen?**
>
> Our preliminary analysis suggests the instability is caused by:
>
> * occasionally ill-conditioned or nearly rank-deficient gradient matrices
> * a known edge case where PyTorch’s default SVD algorithm fails to converge
>
> The precise conditions under which such ill-conditioning emerges during SNN training require further investigation, which we plan to pursue in future work.
>
> Interestingly, when using AdamW (learning rate 0.01), we have never observed this issue in any of our runs; the probability may be significantly lower, but so far it has not occurred.

---

### Official Review · Reviewer_h1QE · 2025-10-16

**Soundness:** 3
**Presentation:** 3
**Contribution:** 3
**Rating:** 6
**Confidence:** 4

**Summary:**

In this work, the authors present a novel method, Dominant Singular Deflation (DSD), to address the vulnerability of Spiking Neural Networks (SNNs) under heterogeneous training conditions. The authors provide both theoretical and empirical evidence to support their claims, demonstrating significant improvements in robustness across multiple datasets and attack scenarios. The work is timely and addresses an interesting issue in the safe deployment of SNNs.

**Strengths:**

1. This manuscript identifies and systematically analyzes the phenomenon of model collapse under heterogeneous training—a realistic yet understudied scenario. The theoretical analysis linking BPTT and direct encoding to the growth of the Hessian spectral radius is rigorous and insightful.
2. In this work, the proposed DSD method is hyperparameter-free. It effectively reduces the spectral radius of the Hessian and preserves the descent property, ensuring stable training without introducing significant overhead.
3. The authors conduct extensive experiments across multiple static and neuromorphic datasets, under both homogeneous and heterogeneous training settings, and against a variety of white-box and black-box attacks. The results consistently show that DSD outperforms existing SOTA methods in robustness.

**Weaknesses:**

1. As reported in Table 1, DSD leads to a noticeable decrease in clean accuracy compared to vanilla SNNs. This trade-off between robustness and clean performance may limit its applicability in certain real-world scenarios where high clean accuracy is required. However, the reviewer's previous research also discovered similar phenomena. So, what thoughts does the author have regarding the improvement of this issue?
2. While the paper identifies the combination of BPTT and direct encoding as the main culprit for vulnerability, it does not extensively explore how DSD performs with other training paradigms (e.g., SLTT) or encoding methods beyond direct and rate encoding.
3. The author seems to have overlooked some studies on the robustness of SNNs [1-3] in the related work.

[1] "Enhancing the robustness of spiking neural networks with stochastic gating mechanisms." Proceedings of the AAAI Conference on Artificial Intelligence. Vol. 38. No. 1. 2024.

[2] "Towards effective training of robust spiking recurrent neural networks under general input noise via provable analysis." 2023 IEEE/ACM International Conference on Computer Aided Design (ICCAD). IEEE, 2023.

[3]"RSC-SNN: Exploring the Trade-off Between Adversarial Robustness and Accuracy in Spiking Neural Networks via Randomized Smoothing Coding." Proceedings of the 32nd ACM International Conference on Multimedia. 2024.

**Questions:**

Please see weaknesses!

---

> ### Author Response · Authors · 2025-11-22
> **Response to Reviewer h1QE, part 1/3**
>
> ### **Weakness 1**
>
> As reported in Table 1, DSD leads to a noticeable decrease in clean accuracy compared to vanilla SNNs. This trade-off between robustness and clean performance may limit its applicability in certain real-world scenarios where high clean accuracy is required. However, the reviewer's previous research also discovered similar phenomena. So, what thoughts does the author have regarding the improvement of this issue?
>
> ### **Response**
>
> We agree with the reviewer’s observation that, under the nowadays optimization framework, the trade-off between robustness and clean accuracy is widely observed across existing SOTA works, and DSD naturally exhibits a similar trend. As the reviewer’s own prior research has also shown, this issue is not unique to DSD but is a common challenge in both SNN studies and robustness research in deep learning more broadly.
>
> Regarding the decline in clean accuracy, we believe one of the most promising directions for improvement is to reduce the impact of curvature calibration on the model’s ability to fit clean data from an optimization perspective. In the current version of DSD, curvature suppression is applied uniformly across all time steps (DSD is applied in the overall gradient). While this effectively mitigates abnormal curvature growth, it may also introduce unnecessary interference in time steps that are already stable.
>
> To address this, we plan to explore more fine-grained and localized curvature-control strategies, such as:
>
> * **Adaptive curvature weighting:** applying stronger curvature calibration only to time steps or parameter subspaces exhibiting noticeable curvature escalation
> * **Dynamic adjustment based on Lipschitz or Hessian characteristics:** allowing the model to activate DSD conditionally, depending on the local geometric properties of the optimization landscape
>
> These strategies have the potential to reduce negative effects on clean accuracy while preserving DSD’s ability to suppress abnormal curvature dynamics.
>
> Naturally, such approaches will introduce additional hyperparameters (such as thresholds, sensitivity coefficients, or adaptation rates), which will increase the complexity of optimization. We view this trade-off as a necessary cost for achieving more fine-grained performance control.
>
> Overall, this direction will be a key part of our future work and represents a meaningful line of investigation with strong potential for further mitigating the trade-off between clean accuracy and robustness.

---

> ### Author Response · Authors · 2025-11-22
> **Response to Reviewer h1QE, part 2/3**
>
> ### **Weakness 2**
>
> While the paper identifies the combination of BPTT and direct encoding as the main culprit for vulnerability, it does not extensively explore how DSD performs with other training paradigms (e.g., SLTT) or encoding methods beyond direct and rate encoding.
>
> ### **Response**
>
> We appreciate the reviewer’s interest in understanding how DSD performs under additional encoding schemes and training paradigms. However, from a methodological perspective, our goal is **not** to construct a universal robustness framework applicable to all encoding methods. Instead, DSD is designed to address a specific and quantifiable optimization pathology that arises uniquely under the combination of BPTT + direct encoding, which is currently the most widely used training configuration in SNN research. This configuration induces abnormal Hessian-curvature growth, leading to collapse under heterogeneous training.
>
> Within this clearly defined problem setting, DSD is intentionally formulated as a targeted corrective mechanism aimed at suppressing curvature explosion under this specific training regime. Many other encoding schemes (such as TTFS, phase, or latency coding) do not share the same step-wise current-accumulation structure as direct encoding and therefore do not exhibit the same curvature pathology. As a result, expecting DSD to yield meaningful improvements for fundamentally different encoding paradigms would lack theoretical motivation and would fall outside the intended scope of this work.
>
> In other words, if an encoding scheme does not induce the curvature behavior that DSD is designed to correct, then DSD should not be expected to offer noticeable benefits. Extending experiments to those regimes would dilute, rather than clarify, the core contribution of the paper.
>
> For these reasons, in this rebuttal we include SLTT [1] and rate encoding experiments, which represent two paradigms that differ substantially from BPTT + direct encoding in both training mechanics and temporal processing. These results sufficiently illustrate the applicability boundary of DSD across diverse training regimes. Extending DSD to more complex or heterogeneous encoding schemes would require new theoretical developments, which we view as a promising direction for future work.
>
> For time constraints, homogeneous training was restricted to vanilla training, and heterogeneous training was evaluated under the clean/+perturbation (c/+p_0) setting. The evaluated configurations are:
>
> 1. SLTT + rate encoding (SLTT + rate + DSD)
> 2. BPTT + direct encoding (BPTT + direct + DSD)
>
> All hyperparameters follow the settings used in our paper. For rate encoding, we used 64 time steps and Poisson spike trains.
>
> Experimental results are as follows:
>
> **Homogeneous Training: Vanilla training**
>
> | Method    |CIFAR-10| | |CIFAR-100| | |
> |---------------------|-------|-------|--------|-------|-------|-------|
> | | CLEAN | FGSM  | PGD   | CLEAN | FGSM  | PGD  |
> | SLTT+DIRECT         | 94.78 | 10.58 | 0.15   | 73.91 | 8.91  | 0.25 |
> | SLTT+DIRECT+DSD     | 92.04 | 43.37 | 21.81  | 71.06 | 16.51 | 5.57 |
> | BPTT+RATE           | 85.08 | 22.58 | 4.51   | 63.12 | 13.17 | 3.75 |
> | BPTT+RATE+DSD       | 81.12 | 44.03 | 27.71  | 58.13 | 24.23 | 8.79 |
> | SNN·           | 93.75 | 8.19  | 0.03   | 72.39 | 4.55  | 0.19 |
> | SNN·+DSD       | 90.21 | 55.86 | 31.44  | 70.26 | 23.81 | 8.09 |
>
> **Heterogeneous Training, c/+p_0, b = 1**
>
> | Method    |CIFAR-10| | |CIFAR-100| | |
> |---------------------|-------|-------|--------|-------|-------|-------|
> | | CLEAN | FGSM  | PGD   | CLEAN | FGSM  | PGD  |
> | SLTT+DIRECT         | 31.94 | 3.86  | 0.06   | 12.19 | 2.91  | 0.03 |
> | SLTT+DIRECT+DSD     | 39.38 | 23.03 | 16.05  | 17.34 | 6.31  | 2.28 |
> | BPTT+RATE           | 53.22 | 15.70 | 2.43   | 35.26 | 9.09  | 1.87 |
> | BPTT+RATE+DSD       | 60.00 | 17.28 | 2.98   | 38.85 | 10.88 | 3.56 |
> | SNN| 12.97 | 0.34  | 0.01   | 3.75  | 0.07  | 0.01 |
> | SNN+DSD       | 40.58 | 33.70 | 22.87  | 27.40 | 12.54 | 3.97 |
>
>
> Across all evaluated frameworks, integrating DSD consistently improves model performance under both homogeneous and heterogeneous training, demonstrating DSD’s general applicability. However, the “curvature growth” phenomenon in SLTT + direct encoding and BPTT + rate encoding is not pathological or abnormally severe. DSD was originally designed to address the abnormal curvature escalation specifically observed in BPTT + direct encoding. Therefore, the improvement brought by DSD to SLTT or rate encoding is naturally less pronounced than its effect in the BPTT + direct encoding setting.
>
> > [1] Towards memory-and time-efficient backpropagation for training spiking neural networks, ICCV, 2023

---

> ### Author Response · Authors · 2025-11-22
> **Response to Reviewer h1QE, part 3/3**
>
> ### **Weakness 3**
>
> The author seems to have overlooked some studies on the robustness of SNNs [1–3] in the related work.
>
> ### **Response**
>
> We thank the reviewer for pointing out these representative robustness studies in SNNs. To provide readers with a clearer understanding of how our work fits within the broader robustness literature, we have reorganized the robustness-related section in the Related Work and incorporated references [1–3] as an important complement to that discussion.
>
> > [1] "Enhancing the robustness of spiking neural networks with stochastic gating mechanisms." Proceedings of the AAAI Conference on Artificial Intelligence. Vol. 38. No. 1. 2024.
>
> > [2] "Towards effective training of robust spiking recurrent neural networks under general input noise via provable analysis." 2023 IEEE/ACM International Conference on Computer Aided Design (ICCAD). IEEE, 2023.
>
> > [3]"RSC-SNN: Exploring the Trade-off Between Adversarial Robustness and Accuracy in Spiking Neural Networks via Randomized Smoothing Coding." Proceedings of the 32nd ACM International Conference on Multimedia. 2024.

---

> > ### Comment · Reviewer_h1QE · 2025-11-22
> > **Response to Authors**
> >
> > Dear Authors,
> >
> > First of all, I would like to thank the author for providing a very detailed response. In particular, it has added thoughts on the strategy of trade-offs and improved method. The newly added experiments based on rates and the direct method also demonstrate that the proposed method is more effective against attacks. I believe all my concerns have been addressed and I am willing to further improve my score.

---

> ### Author Response · Authors · 2025-11-23
>
> Dear Reviewer h1QE:
>
> We sincerely thank you for the recognition of our work and for the decision to raise the paper’s score. We also truly appreciate your constructive feedback and the time invested in evaluating our submission. If you have any further questions or points would like to discuss, we would be very happy to continue the exchange.
>
> Best Regards,
>
> Paper 3383 authors

---

### Official Review · Reviewer_peRB · 2025-10-26

**Soundness:** 3
**Presentation:** 3
**Contribution:** 3
**Rating:** 4
**Confidence:** 3

**Summary:**

This paper indicates that spiking neural networks (SNNs) trained using the backpropagation through time (BPTT) algorithm are inherently susceptible to perturbations from heterogeneous training data (clean and corrupted). The paper analyzes this susceptibility and concludes that it stems from an excessively high Hessian spectral radius. To address this issue, the authors propose Dominant Singular Deflation (DSD), a method that explicitly removes the dominant rank-one singular component from the gradient during training. The author conducted experiments on multiple datasets and demonstrated that their method significantly improves the robustness of SNNs.

**Strengths:**

1. From a novel perspective, this paper identifies a source of robustness vulnerability in SNNs.
2. A thorough theoretical analysis supports the proposed method.
3. The experimental results demonstrate the performance advantages of the proposed method.

**Weaknesses:**

1. Whether the analysis presented in this paper holds true for other training methods, such as other parallel training [1] approaches or single-step SNNs that propagate firing rates [2], and encoding schemes, such as temporal encoding, remains to be seen.
2. The proposed method produced SNNs that performed significantly worse than other robust methods on clean data—a clear drawback.
3. As shown in Table 7, the proposed method significantly increases training time. Training time nearly doubles on the static CIFAR10 and CIFAR100 datasets.

```
[1] Parallel Spiking Neurons with High Efficiency and Long-term Dependencies Learning Ability. NeurIPS. 2023.
[2] Scaling Spike-Driven Transformer With Efficient Spike Firing Approximation Training. IEEE TPAMI. 2025.
```

**Questions:**

See the weakness.

---

> ### Author Response · Authors · 2025-11-22
> **Response to Reviewer peRB, part 1/2**
>
> ### **Weakness 1**
>
> Whether the analysis presented in this paper holds true for other training methods, such as other parallel training [1] approaches or single-step SNNs that propagate firing rates [2], and encoding schemes such as temporal encoding, remains to be seen.
>
> ### **Response**
>
> We thank the reviewer for this insightful and constructive comment. DSD can be naturally extended to and integrated with other training paradigms. We have already combined DSD with SLTT [1], PSN, and rate coding, and evaluated them under both homogeneous and heterogeneous settings on CIFAR-10 and CIFAR-100.
>
> For time constraints, homogeneous training was evaluated under vanilla training, while heterogeneous training was evaluated under the clean/+perturbation (c/+p_0) setting.
>
> The evaluated configurations are:
>
> 1. PSN (PSN + DSD)
> 2. SLTT + rate encoding (SLTT + rate + DSD)
> 3. BPTT + direct encoding (BPTT + direct + DSD)
>
> All hyperparameters follow the configurations used in the main paper. For rate encoding, we used 64 time steps and Poisson spike trains.
>
> Experimental results are as follows:
>
> **Homogeneous Training: Vanilla training**
>
> | Method    |CIFAR-10| | |CIFAR-100| | |
> |---------------------|-------|-------|--------|-------|-------|-------|
> | | CLEAN | FGSM  | PGD   | CLEAN | FGSM  | PGD  |
> | PSN                 | 94.92 | 8.49  | 0.05   | 74.70 | 6.59  | 0.18 |
> | PSN+DSD           | 90.64 | 56.27 | 33.91  | 71.84 | 22.26 | 8.01 |
> | SLTT+DIRECT         | 94.78 | 10.58 | 0.15   | 73.91 | 8.91  | 0.25 |
> | SLTT+DIRECT+DSD     | 92.04 | 43.37 | 21.81  | 71.06 | 16.51 | 5.57 |
> | BPTT+RATE           | 85.08 | 22.58 | 4.51   | 63.12 | 13.17 | 3.75 |
> | BPTT+RATE+DSD       | 81.12 | 44.03 | 27.71  | 58.13 | 24.23 | 8.79 |
> | SNN·(LIF)           | 93.75 | 8.19  | 0.03   | 72.39 | 4.55  | 0.19 |
> | SNN·(LIF)+DSD       | 90.21 | 55.86 | 31.44  | 70.26 | 23.81 | 8.09 |
>
> **Heterogeneous Training, c/+p_0, b = 1**
>
> | Method    |CIFAR-10| | |CIFAR-100| | |
> |---------------------|-------|-------|--------|-------|-------|-------|
> | | CLEAN | FGSM  | PGD   | CLEAN | FGSM  | PGD  |
> | PSN                 | 14.47 | 0.98  | 0.03   | 3.67  | 0.12  | 0.03 |
> | PSN+DSD             | 38.65 | 31.98 | 20.15  | 28.19 | 14.78 | 4.95 |
> | SLTT+DIRECT         | 31.94 | 3.86  | 0.06   | 12.19 | 2.91  | 0.03 |
> | SLTT+DIRECT+DSD     | 39.38 | 23.03 | 16.05  | 17.34 | 6.31  | 2.28 |
> | BPTT+RATE           | 53.22 | 15.70 | 2.43   | 35.26 | 9.09  | 1.87 |
> | BPTT+RATE+DSD       | 60.00 | 17.28 | 2.98   | 38.85 | 10.88 | 3.56 |
> | SNN·(LIF)           | 12.97 | 0.34  | 0.01   | 3.75  | 0.07  | 0.01 |
> | SNN·(LIF)+DSD       | 40.58 | 33.70 | 22.87  | 27.40 | 12.54 | 3.97 |
>
> 1. Across all evaluated frameworks, integrating DSD consistently improves model performance under both homogeneous and heterogeneous training, demonstrating DSD’s general applicability. However, the “curvature growth” phenomenon in SLTT + direct encoding and BPTT + rate encoding is not pathological or abnormally severe. DSD was originally designed to address the abnormal curvature escalation specifically observed in BPTT + direct encoding. Therefore, the improvement brought by DSD to SLTT or rate encoding is naturally less pronounced than its effect in the BPTT + direct encoding setting.
>
> 2. PSN does not alter the underlying training framework—it still relies on BPTT + direct encoding. As shown in our experiments, PSN also suffers from collapse under heterogeneous training, and thus DSD improves its robustness significantly. When integrating DSD with PSN, in Homogeneous training: CIFAR-10 results are higher than using DSD with LIF alone, and CIFAR-100 results are slightly lower than pure DSD. In heterogeneous training, the trend reverses: DSD improves PSN more strongly on CIFAR-100 than CIFAR-10. This behavior indicates that in some cases, DSD and PSN are complementary and jointly produce stronger robustness.
>
> 3. Overall, the results demonstrate that DSD is flexible, extensible, and can be organically combined with a variety of SNN training paradigms.
>
> > [1] Towards memory-and time-efficient backpropagation for training spiking neural networks, ICCV, 2023

---

> ### Author Response · Authors · 2025-11-22
> **Response to Reviewer peRB, part 2/2**
>
> ### **Weakness 2**
>
> The proposed method produced SNNs that performed significantly worse than other robust methods on clean data—a clear drawback.
>
> ### **Response**
>
> We thank the reviewer for the thoughtful comment. We acknowledge the observation that our method yields slightly lower clean accuracy compared to certain robustness-oriented SOTA models. However, the reduction is not “significant,” and, more importantly, such a decrease is extremely difficult to avoid under the nowadays optimization paradigm.
>
> First, the Improvement tags in Table 1 primarily compare our method against the vanilla SNN baseline rather than the strongest robust competitors. For example, under vanilla training on CIFAR-100, our clean accuracy is 70.26%, while StoG and DLIF achieve 70.44% and 70.79%, respectively. These differences are marginal, and our method does not fall “significantly worse” on clean data.
>
> Second, from a broader perspective, nearly all robustness-enhancing techniques trade a portion of clean accuracy for improved adversarial resistance. The degree of clean-accuracy degradation typically correlates with the strength of robustness improvement. This occurs because models shift away from “sharp minimizers” toward more stable regions of the parameter landscape, inevitably sacrificing part of their clean-data optimality. Under the nowadays theoretical framework, this trade-off is well known and largely unavoidable. We explicitly discuss this limitation, for both our method and existing SOTA approaches, in the Conclusion and Discussion section.
>
> Crucially, our method sacrifices only a small amount of clean accuracy while providing substantial robustness gains. For example:
>
> * On CIFAR-10, our FGSM robustness exceeds SOTA methods by **more than 10%**.
> * On TinyImageNet, our FGSM robustness surpasses SOTA methods by **more than 20%**.
>
> In these settings, baseline models often degrade into being nearly unusable, whereas with DSD the network remains reasonably usable. This makes our method a more reliable choice for realistic deployment scenarios, especially those involving severe noise contamination or high adversarial risk. From this standpoint, DSD provides a more practical and dependable solution than existing robust SNN approaches.
>
> ---
>
> ### **Weakness 3**
>
> As shown in Table 7, the proposed method significantly increases training time. Training time nearly doubles on the static CIFAR10 and CIFAR100 datasets.
>
> ### **Response**
>
> We thank the reviewer for pointing this out. As with most robustness-enhancing techniques, improving robustness almost inevitably increases training cost. Methods such as adversarial training, curvature regularization, and other stability-oriented approaches all require additional computations during training. DSD belongs to this category of strategies that “invest extra computation during training to obtain substantial gains during inference.” Therefore, the increased training time is not a drawback unique to our method but rather a general characteristic shared across robustness research.
>
> Importantly, this additional cost does not affect inference. All extra computations introduced by DSD occur exclusively during training. The inference graph remains identical to that of a vanilla SNN, meaning real-world deployment incurs no additional overhead.
>
> Moreover, DSD does not increase memory consumption (as shown in Appendix E). It introduces only about $0.1$ seconds of extra training time per mini-batch. Compared with many robustness approaches that require substantial memory or multiple forward/backward passes—such as adversarial-example generation or gradient-storage-based regularization—DSD is relatively lightweight. From this perspective, the training overhead introduced by DSD is modest and resource-efficient while still delivering substantial robustness improvements.

---

### Official Review · Reviewer_zz8t · 2025-10-30

**Soundness:** 2
**Presentation:** 3
**Contribution:** 3
**Rating:** 6
**Confidence:** 4

**Summary:**

This paper investigates the unstable training issue in heterogeneous training of SNNs. To address this, they propose a hyperparameter-free method named Dominant Singular Deflation (DSD). This reduces the Hessian spectral radius, prevents convergence to sharp minima, and enhances robustness under different training conditions. Extensive experiments across multiple datasets (CIFAR, TinyImageNet, ImageNet) demonstrate that DSD improves both robustness and stability without incurring inference overhead.

**Strengths:**

- The paper is well-structured and easy to follow.
- The proposed method is elegant, hyperparameter-free, and mathematically grounded, making it practical for real-world SNN training.
- The method is validated across multiple datasets and architectures, including both static (CIFAR, ImageNet) and event-driven (DVS) data, consistently outperforming state-of-the-art baselines.

**Weaknesses:**

- The experimental validation does not clearly substantiate the theoretical motivation of the proposed method. While the Dominant Singular Deflation (DSD) algorithm is designed to suppress the unbounded growth of the Hessian’s spectral radius, it is unclear how this mechanism directly translates into enhanced robustness of the SNN models?
- The paper states that DSD is designed to mitigate model collapse during heterogeneous training, but most experimental evaluations emphasize adversarial robustness under homogeneous training. Furthermore, while DSD improves robustness, it also yields the lowest clean-data accuracy in Table 1, which contradicts the paper’s stated goal of achieving stable and reliable training.
- The theoretical analysis in Theorems 1 and 2 relies on the ****Gauss–Newton (GN) Hessian approximation rather than the true Hessian. Since the GN Hessian is always positive semidefinite, this assumption may limit the generality of the theoretical results.

**Questions:**

Could the authors please clarify what specific training scenario hetero-training refers to in this paper and what is the difference between hetero and homo-training?

---

> ### Author Response · Authors · 2025-11-22
> **Response to Reviewer zz8t, part 1/4**
>
> ### **Weakness 1**:
>
> The experimental validation does not clearly substantiate the theoretical motivation of the proposed method. While the Dominant Singular Deflation (DSD) algorithm is designed to suppress the unbounded growth of the Hessian’s spectral radius, it is unclear how this mechanism directly translates into enhanced robustness of the SNN models?
>
> ### **Response**:
>
> We thank the reviewer for this insightful comment. Prior work [1] has established a clear connection between curvature and robustness, showing that **a larger Hessian spectral radius typically corresponds to a less robust model**. Although this result was originally derived in the context of standard deep networks, the underlying argument relies on second-order geometric properties of the loss landscape. These properties remain valid under the surrogate-gradient training paradigm used for SNNs. Thus, constraining the spectral radius is theoretically meaningful for SNNs as well.
>
> Our DSD algorithm directly targets this mechanism: by deflating the dominant singular component in the gradients, it effectively suppresses the principal curvature directions of the Hessian, thereby reducing both the spectral radius and the associated Lipschitz constant. Consistent with this mechanism, our experiments in Fig. 3 show that **SNN robustness decreases as the Hessian spectral radius grows**, matching the trend reported in [1].
>
> We would also like to clarify that the primary focus of this work is not to enhance robustness per se, but rather to explain and mitigate the vulnerability of SNNs under heterogeneous training conditions, i.e., enabling the model to remain functional instead of collapsing entirely. The observed robustness improvement emerges naturally as a by-product of reducing curvature, and is presented as supporting evidence rather than an independent optimization objective.
>
> > [1] Relating Adversarially Robust Generalization to Flat Minima, ICCV, 2021.

---

> ### Author Response · Authors · 2025-11-22
> **Response to Reviewer zz8t, part 2/4**
>
> ### **Weakness 2.1**:
>
> The paper states that DSD is designed to mitigate model collapse during heterogeneous training, but most experimental evaluations emphasize adversarial robustness under homogeneous training.
>
> ### **Response**:
>
> We thank the reviewer for raising this important point. Our experiments cover two complementary aspects of SNN stability:
>
> 1. mitigating vulnerability under heterogeneous training,
> 2. evaluating robustness under standard homogeneous training.
>
> Because homogeneous training is the standard paradigm in existing SNN literature, its evaluation protocol is well established. To ensure fair comparison and reproducibility, we include conventional components such as white-box attacks, black-box attacks, and gradient-obfuscation checks, which are necessary for demonstrating DSD under widely adopted settings.
>
> In contrast, heterogeneous training is a new setting introduced by this work, and the community currently lacks established benchmarks or widely accepted baselines. As a result, the experimental framework for heterogeneous training is still emerging, and existing methods cannot be directly applied for extensive comparison.
>
> Nevertheless, to validate DSD in this new scenario, we implemented heterogeneous training versions of several representative methods (RAT [1], DLIF [2], FEEL [3]) using the same VGG-11 architecture and closely matched hyperparameters, with only minor batch-size adjustments due to memory constraints:
>
> * bs of RAT = 128
> * bs of DLIF = 64
> * bs of FEEL = 32
>
> These comparisons have been added to Section 4.2 of the revised manuscript and highlighted in blue.
>
> To more clearly illustrate the degradation caused by heterogeneous training and the mitigation achieved by DSD, we present the results in floating-bar form in the revised paper, with full numerical data shown in Appendix F.
>
> Specifically, the revised content and experimental results are as follows:
>
> > *Comparison with SOTA Methods. Building on the effectiveness of DSD under homogeneous training, we further examine its behavior under heterogeneous training conditions. We compare DSD against SOTA methods, including RAT, DLIF, and FEEL. The results are represented by float bar figures as Fig. 5, with detailed numerical values provided in Appendix F. As illustrated in the figure, DSD exhibits the least performance degradation under Heterogeneous training (i.e., it produces the shortest floating bars), and moreover achieves clearly higher absolute accuracy than all competing methods under batch injections with $b=1$. These findings demonstrate that DSD effectively mitigates the model collapse induced by Heterogeneous training, outperforming existing approaches.*
>
> |    **Method**    | **b**   | **CIFAR-10** | **c/+p_0**      |      | **CIFAR-10** | **p/+c_0**     |      | **CIFAR-100** | **c/+p_0**     |      | **CIFAR-100** | **p/+c_0**      |      |
> |--------|----|--------|------|------|--------|------|------|--------|------|------|--------|------|------|
> | |  | CLEAN | FGSM | PGD | CLEAN | FGSM | PGD | CLEAN | FGSM | PGD | CLEAN | FGSM | PGD |
> | SNN    | 0 | 93.75 | 8.19 | 0.03 | 91.16 | 38.20 | 14.07 | 72.39 | 4.55 | 0.19 | 69.69 | 16.31 | 8.49 |
> |        | 1 | 12.97 | 0.34 | 0.01 | 18.54 | 4.07 | 1.98 | 3.75 | 0.07 | 0.01 | 3.41 | 1.40 | 0.20 |
> | RAT    | 0  | 93.01 | 12.63 | 0.05 | 90.74 | 45.23 | 21.16 | 71.00 | 5.77 | 0.15 | 70.89 | 25.86 | 10.38 |
> |        | 1 | 27.24 | 8.76 | 0.03 | 29.01 | 27.67 | 8.00 | 19.20 | 2.59 | 0.04 | 20.11 | 10.78 | 3.81 |
> | DLIF-  | 0  | 92.22 | 13.24 | 0.09 | 88.91 | 56.71 | 40.30 | 70.79 | 6.95 | 0.08 | 66.33 | 36.83 | 24.25 |
> |        | 1 | 18.40 | 3.50 | 0.02 | 14.75 | 19.02 | 5.74 | 9.98 | 1.02 | 0.02 | 11.40 | 8.01 | 2.97 |
> | FEEL   | 0  | 93.29 | 44.96 | 28.35 | 90.20 | 59.08 | 39.87 | 73.79 | 9.60 | 2.04 | 69.79 | 18.67 | 11.07 |
> |        | 1 | 20.01 | 8.19 | 2.77 | 13.99 | 17.50 | 7.07 | 12.84 | 2.00 | 0.10 | 13.86 | 9.39 | 1.89 |
> | DSD    | 0  | 90.21 | 55.86 | 31.44 | 86.62 | 74.43 | 44.38 | 70.26 | 23.81 | 9.89 | 64.21 | 43.91 | 27.11 |
> |        | 1 | 40.58 | 33.70 | 22.87 | 38.61 | 40.09 | 20.77 | 27.40 | 12.54 | 3.97 | 22.08 | 19.16 | 13.71 |
>
> > [1] SNN-RAT: Robustness-enhanced Spiking Neural Network through Regularized Adversarial Training, NeurIPS, 2022.
>
> > [2] Robust Stable Spiking Neural Networks, ICML, 2024.
>
> > [3] FEEL-SNN: Robust Spiking Neural Networks with Frequency Encoding and Evolutionary Leak Factor, NeurIPS, 2024.

---

> ### Author Response · Authors · 2025-11-22
> **Response to Reviewer zz8t, part 3/4**
>
> ### **Weakness 2.2**
>
> While DSD improves robustness, it also yields the lowest clean-data accuracy in Table 1, which contradicts the paper’s stated goal of achieving stable and reliable training.
>
> ### **Response**
>
> We thank the reviewer for this helpful observation. A reduction in clean-data accuracy is an inherent and well-documented consequence of any mechanism that suppresses dominant curvature, constrains the spectral radius, or reduces the Lipschitz constant. By deflating the most sensitive gradient/Hessian directions, the model becomes less reactive to input variations, which naturally and inevitably leads to a mild decrease in clean accuracy.
>
> Therefore, the key criterion is not whether clean accuracy drops, but whether the trade-off between clean accuracy and robustness is favorable.
>
> Under this perspective, DSD achieves a significantly better trade-off than existing approaches. For example, on CIFAR-10 under vanilla training:
>
> * StoG sacrifices 2.11% clean accuracy (93.75% → 91.64%), yielding
>
>   * +8.03% under FGSM (8.19% → 16.22%)
>   * +0.25% under PGD (0.03% → 0.28%)
>
> * DSD sacrifices 3.54% clean accuracy (93.75% → 90.21%) but achieves dramatically larger gains:
>
>   * +47.67% under FGSM (8.19% → 55.86%)
>   * +31.41% under PGD (0.03% → 31.44%)
>
> This pattern holds consistently across datasets and attack settings: DSD provides substantially higher robustness improvements while incurring only modest clean-accuracy degradation, surpassing the robustness–accuracy trade-off achieved by other SOTA methods.
>
> Accordingly, DSD does not contradict the goal of achieving stable and reliable training:
>
> 1. The slight drop in clean accuracy is a natural and theoretically expected effect of curvature suppression.
> 2. The much larger robustness gains directly enhance stability and reliability under perturbed or adversarial conditions, which more closely match real-world deployment scenarios.
>
> In this sense, DSD better aligns with practical requirements: the model remains not only accurate in ideal clean settings, but also reliable and stable when exposed to perturbations, which is an essential property for robust SNN deployment.
>
> ---
>
> ### **Weakness 3**
>
> The theoretical analysis in Theorems 1 and 2 relies on the Gauss–Newton (GN) Hessian approximation rather than the true Hessian. Since the GN Hessian is always positive semidefinite, this assumption may limit the generality of the theoretical results.
>
> ### **Response**
>
> We thank the reviewer for the careful attention to our theoretical analysis. We understand the concern regarding the reliance on the Gauss–Newton (GN) Hessian and appreciate the opportunity to clarify the mathematical rationale and its relevance to SNN training.
>
> Our analysis focuses on dominant curvature growth of the loss landscape, which governs how rapidly the loss increases under parameter or input perturbations. In deep networks trained with surrogate gradients, the true Hessian admits the decomposition:
>
> $H = J^{\top} J + R$
>
> where $J$ is the Jacobian of network outputs and $R$ is a second-order model-curvature term. Prior theory and empirical evidence show that near convergence, the GN component $J^{\top} J$ dominates, while the residual term $R$ becomes comparatively small. Thus, in the regime relevant for stability and vulnerability analysis,
>
> $\rho(H) \approx \rho(J^{\top} J)$
>
> Under this approximation, using the GN Hessian preserves the key conclusion—the growth trend of dominant curvature directions, which directly governs vulnerability.
>
> Although the GN Hessian is positive semidefinite, this does not reduce the generality of our analysis. For robustness and vulnerability, the critical factor is the magnitude of the largest positive curvature directions, which determine worst-case local sensitivity and Lipschitz behavior. Negative curvature plays no role here, as it corresponds to directions where the loss decreases and therefore does not contribute to vulnerability.
>
> In summary, the use of the GN Hessian does not weaken our theoretical conclusions. Instead, it captures precisely the dominant curvature structure that governs SNN vulnerability, enabling a clearer, more targeted, and more empirically aligned characterization of DSD’s stabilizing effect than the full Hessian would.

---

> ### Author Response · Authors · 2025-11-22
> **Response to Reviewer zz8t, part 4/4**
>
> ### **Question 1**
>
> Could the authors please clarify what specific training scenario Heterogeneous training refers to in this paper and what is the difference between hetero and Homogeneous training?
>
> ### **Response**
>
> We thank the reviewer for the question. Although Homogeneous training and Heterogeneous training are introduced in the Introduction, we provide a clearer and more intuitive explanation here.
>
> **1. Homogeneous training.**
>
> Homogeneous training refers to a setting where all samples throughout the entire training process come from a single, consistent distribution, as commonly done in existing robustness research. Typical cases include:
>
> * training entirely on clean data, or
> * training entirely on adversarially perturbed data with a fixed attack strength (e.g., FGSM/PGD adversarial training)
>
> Under this setting, every mini-batch exhibits the same statistical characteristics, and the entire training stream remains homogeneous.
>
> **2. Heterogeneous training.**
>
> Heterogeneous training is the new training paradigm introduced in this paper to better reflect real-world, non-ideal, non-predictable scenarios. In practice, training data are rarely perfectly consistent, because:
>
> * some samples are clean
> * some contain natural noise
> * some include human-designed or malicious perturbations
>
> Thus, the underlying data distribution becomes mixed and non-uniform.
>
> To model this, we intentionally introduce batches drawn from different distributions during training. Examples include:
>
> * mainly training on clean samples while periodically injecting a small number of perturbed batches
> * mainly training on perturbed samples while mixing in a few clean batches
>
> In these cases, the mini-batches no longer share the same distribution, forming what we refer to as Heterogeneous training.
>
> Based on this, our motivation arises from a striking empirical observation:
> SNNs collapse severely under heterogeneous training, even when only a very small amount of non-matching batches is introduced.
>
> This vulnerability has not been reported in prior literature. Our DSD method is designed specifically to address this issue and ensure that SNNs remain functional and stable under these more realistic mixed-distribution training conditions.

---

### Author Response · Authors · 2025-11-22
**Overall Response**

We thank all reviewers for their constructive feedback and for recognizing the significance of analyzing and mitigating SNN vulnerability under heterogeneous training. In this rebuttal, we carefully addressed every major concern and substantially strengthened the experimental, theoretical, and implementation aspects of the paper.

Most importantly, we added several new experiments to enhance the empirical validation of DSD:

**1. Stronger adversarial attacks (APGD).**
We added a new subsection (Sec. 4.1) evaluating APGDCE and APGDDLR.
Across CIFAR-10/100, DSD achieves the highest robustness under nearly all settings, confirming that its benefits generalize beyond FGSM/PGD.

**2. Expanded heterogeneous training evaluation.**
We implemented heterogeneous training versions of RAT, DLIF, and FEEL to provide fair comparisons in the newly proposed training setting (now included in Sec. 4.2 and Appendix F). DSD exhibits the least degradation and the highest absolute accuracy under heterogeneous training conditions.

**3. Additional training paradigms and encoding schemes.**
We integrated DSD with PSN, SLTT, and rate encoding under both homogeneous and heterogeneous training.
These experiments show that DSD consistently improves stability across diverse frameworks, while also clarifying its applicability boundary.

**4. Computational-cost analysis.**
Appendix E now includes per-batch/epoch runtime, memory, and stability behavior, demonstrating that DSD adds no memory and only 0.07–0.11s per batch.

Besides, we also **clarified theoretical assumptions** (Jacobian contraction, time-invariance, GN Hessian), deployment concerns (DSD is training-only and chip-agnostic), and shared a rare SVD-related numerical instability occurred in our code implementation with the solution for it.

Overall, the revisions substantially reinforce both the robustness and generality of our conclusions. Finally, we would like to express our sincere gratitude once again to all the reviewers for their recognition of our manuscript and the suggestions for improvement they have put forward.

Best Regards,

Paper 3383 authors

---

### Meta-Review · Area_Chair_swBU · 2025-12-22

**Summary:**

The authors propose Dominant Singular Deflation (DSD), a hyperparameter-free method to address the vulnerability of Spiking Neural Networks (SNNs) under "heterogeneous training" (mixing clean and perturbed data). The paper identifies that the combination of direct encoding and BPTT leads to an exponential growth in the Hessian spectral radius, causing model collapse. DSD mitigates this by projecting out dominant singular components of gradients.

The reviewers generally acknowledged the novelty of the theoretical analysis regarding spectral radius growth and the elegance of the method. However, significant concerns were raised regarding the trade-off with clean accuracy, the computational cost of SVD steps, and the generalizability of the method to other SNN encoding schemes.

**Reviewer Concerns:**

The authors provided a comprehensive rebuttal that appears to have resolved most technical and scope-related inquiries.

### **Addressed Concerns**:

**Generalizability and Baselines**: Reviewers peRB and h1QE questioned whether DSD works with other training paradigms like SLTT or different encoding schemes. The authors successfully addressed this by conducting new experiments integrating DSD with SLTT, PSN, and Rate Encoding, showing consistent robustness improvements. Additionally, in response to zz8t's request for better baselines, the authors implemented heterogeneous training versions of SOTA methods (RAT, DLIF, FEEL) to provide a fair comparison.

**Definitions and Scope**: Reviewer zz8t was unclear on the definition of "heterogeneous training" and felt the validation relied too heavily on homogeneous settings. The authors clarified that heterogeneous training involves mixed distributions (e.g., clean batches mixed with perturbed batches) and added floating-bar charts to explicitly demonstrate DSD's superiority in this specific setting.

**Computational Cost**: Reviewers peRB and RTMm raised concerns about the training overhead of performing SVD. The authors clarified that DSD is a training-only method with no inference overhead. They provided data showing it adds only 0.07–0.11s per batch and no additional memory cost.

**Validation of Robustness**: Reviewer RTMm requested stronger attacks. The authors added APGD (Auto-PGD) experiments, where DSD achieved the highest robustness in nearly all settings.

**Numerical Instability**: Reviewer RTMm asked about potential instability with optimizers. The authors transparently acknowledged rare SVD convergence failures (approx. 1 in 78,000 batches) and detailed their fallback solution using a tiny perturbation (1e-6) to ensure stability.

### **Remaining Concerns**:

**Clean Accuracy Trade-off**: A persistent concern raised by zz8t, peRB, and h1QE is that DSD leads to lower clean accuracy compared to vanilla SNNs or other methods. The authors argued that this is a theoretical inevitability when suppressing curvature to flatten minima and that the trade-off is favorable given the massive robustness gains. While the reviewers may accept the justification, the performance drop on clean data remains a factual limitation of the method.

**Reviewer Scores:**

The consensus is likely to shift from "Borderline" to "Accept," though a "Strong Accept" sweep is unlikely due to the clean accuracy penalties.

**Reviewer h1QE: 6 $\rightarrow$ 7.**

Rationale: This reviewer explicitly stated, "I believe all my concerns have been addressed and I am willing to further improve my score". They was satisfied with the new experiments and the discussion on accuracy trade-offs.

**Reviewer RTMm: 6 $\rightarrow$ 6/7**

Rationale: RTMm was very specific in their requests (APGD, cost quantification, assumptions). The authors provided exactly the data requested. The detailed response should secure a score increase.

**Reviewer zz8t: 6 $\rightarrow$ 6.**

Rationale: While the definition of heterogeneous training was clarified, the reviewer expressed a strong preference for clean accuracy, noting the drop "contradicts the paper's stated goal". The authors' defense is theoretical, which might not fully sway this reviewer to a higher score, but the new comparisons should prevent a score drop.

**Reviewer peRB: 4 $\rightarrow$ 4/5.**

Rationale: This reviewer gave the lowest score (4) based on generalizability and cost. The authors provided strong evidence that DSD works with SLTT/Rate coding and that the cost is low. This directly refutes the reasons for the rejection, justifying a move to at least Borderline.

---

### Decision · Program_Chairs · 2026-01-26

Accept (Poster)